# SAMPLE MARGIN-AWARE RECALIBRATION OF TEMPERATURE

## ABSTRACT

Deep neural networks often exhibit overconfidence despite their high accuracy. Such miscalibration limits reliability in safety-critical domains where trustworthiness are crucial. Post-hoc calibration methods offer a practical solution where popular approaches like Temperature Scaling (TS) apply a single corrective parameter to all samples, failing to address the sample-dependent nature of miscalibration. While more advanced methods attempt to adapt to sample difficulty, they often rely on complex and indirectly learned proxies. In this work, we first identify the *logit margin* as a direct, simple, and principled indicator of sample hardness. We provide substantial empirical and theoretical evidence that it serves as a more effective indicator of sample hardness than existing proxies. Meanwhile, we identify a fundamental flaw in current methods that optimizing Negative Log-Likelihood (NLL) can paradoxically degrade calibration. To resolve this, we introduce Charbonnier–SoftECE, a novel and theoretically guaranteed objective that directly minimizes calibration error. Building on these insights, we propose Sample Margin-Aware Recalibration of Temperature (SMART), a lightweight post-hoc method that learns a minimalistic sample-wise mapping from the logit margin to an optimal temperature, guided by our calibration-centric objective. Extensive experiments show state-of-the-art performance for calibration across diverse architectures and datasets with a minimal inference-time data consumption. The code is available at: `https://anonymous.4open.science/r/SMART-8B11`.

## 1 INTRODUCTION

Deep neural networks have achieved remarkable success across diverse domains, yet their deployment in safety-critical applications such as autonomous driving Feng et al. (2019) and medical diagnosis Chen et al. (2018) demands more than just high predictive accuracy. These high-stakes scenarios require models to provide reliable uncertainty estimates that accurately reflect the true likelihood of prediction correctness, i.e., *calibration* Guo et al. (2017). A well-calibrated model ensures informed decision-making and appropriate deferral to human experts when uncertainty is high. However, current models commonly suffer from severe miscalibration Guo et al. (2017), primarily overconfidence Guo et al. (2017); Wei et al. (2022); Luo et al. (2025), where models assign high confidence scores to predictions that are frequently incorrect. The real-world consequences of such overconfident behavior can be catastrophic, such as wrong diagnostic decisions with high confidence.

To address miscalibration, the research community has developed two primary streams of solutions. *Train-time calibration* methods integrate calibration directly into the learning process via specialized data Wang et al. (2023); Hendrycks et al. (2020), training framework Tao et al. (2023), regularizations Müller et al. (2019); Pereyra et al. (2017), and designed loss objectives Mukhoti et al. (2020); Tao et al. (2023). However, these methods hardly apply to trained models. In contrast, *post-hoc calibration* methods Zadrozny & Elkan (2002; 2001) operate easily on large pretrained models. Due to its simplicity and effectiveness, Temperature Scaling (TS) Guo et al. (2017) has become the most widespread post-hoc method that learns a single scaling value on the validation set.However, this one-size-fits-all approach is inherently problematic, as miscalibration is not uniform across samples. To address this, several methods have been proposed to learn separate temperatures per class Frenkel & Goldberger (2021) or semantic-aware groupings through clustering Yang et al. (2024). To facilitete more fine-grained temperature scaling, sample-adaptive methods propose to operate on distinctive sample-wise information Ding et al. (2021); Tomani et al. (2022).

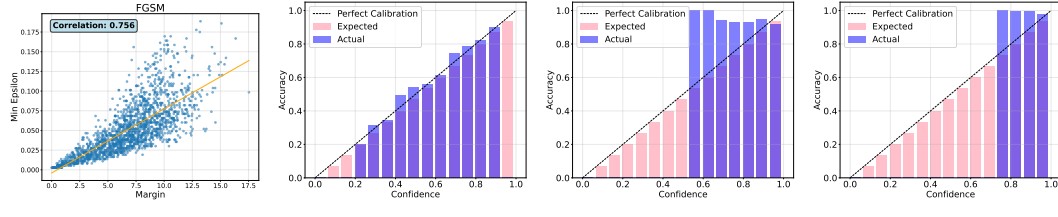

(a) Attack vs Margin   (b) Margin range 1.78-2.90   (c) Margin range 4.00-4.99   (d) Margin range 6.76-8.68

Figure 1: Relationship between min perturbation of FGSM Goodfellow et al. (2015) and logit margin on CIFAR-10, with reliability diagrams for various margin groups from left to right.

Xiong et al. (2024) calibrate predictions based on sample proximity, assigning larger temperatures to less proximate samples; Yang et al. (2024) apply larger temperatures to groups that are harder to distinguish (e.g., birds and airplanes sharing the same background in CIFAR-10); and Ding et al. (2021) exploit feature-space sparsity to adaptively guide temperature. Despite their methodological differences, these approaches share the same underlying motivation: sample hardness drives calibration. However, they rely on *indirectly learned* proxies of difficulty. In contrast, we propose a direct and simple measure—the *logit margin*, defined as the gap between the largest and second-largest logits. Empirical results (Figure 1b–d) show that larger-margin samples are systematically easier and more under-confident, even when their confidence levels are identical. Moreover, the strong correlation between the margin and the minimum perturbation required to reach the decision boundary under attack (Figure 1a) highlights the margin's reliability as a hardness indicator. Finally, our theoretical analysis in Appendix A.1 demonstrates that the optimal temperature for any target confidence is tightly bounded by the margin, underscoring its effectiveness as a principled signal of sample difficulty for post-hoc calibration.

Another limitation inherent in current scaling-based methods is that they focus on optimizing the NLL loss, which theoretically does not guarantee a reduction in the calibration errors. In fact, as we prove in Appendix A.3, certain scenarios can lead to a paradoxical outcome where NLL decreases while ECE simultaneously increases, thereby defeating the primary goal of calibration. To address this fundamental misalignment, we adopt a novel scaling objective function, Charbonnier–SoftECE. This new objective directly targets the calibration error. As also established by our theoretical analysis in Appendix A.2, optimizing with Charbonnier–SoftECE provably resolves the issue inherent in the NLL loss, ensuring that the optimization process aligns directly with the goal of improving calibration.

Building on these validated insights, we introduce **S**ample **M**argin-**A**ware **R**ecalibration of **T**emperature (SMART), a lightweight post-hoc calibration method that aims to learn a direct and minimalistic mapping from logit margins to temperatures: $T(\cdot) : \mathbb{R}^+ \to \mathbb{R}^+$. Using Charbonnier–SoftECE as its learning objective, SMART is theoretically guaranteed to yield superior calibration. Experiments on various benchmarks and architectures validate the state-of-the-art effectiveness and efficiency of SMART, even with a minimal validation set.

**Contributions** Our work is theory-driven and makes three key contributions: we first provide formal and empirical analysis showing that logit margin is a principled hardness indicator that tightly bounds the feasible temperature range, outperforming existing proxies with minimal computation; second, we prove a fundamental mismatch between NLL optimization and calibration quality, and resolve it through a novel Charbonnier–SoftECE objective that provably upper-bounds smooth calibration error; finally, building on these theoretical insights, we develop SMART, a lightweight margin-aware temperature mapping that achieves state-of-the-art calibration on CNNs and ViTs across long-tail and out-of-distribution datasets, remaining effective with as few as 50 validation samples.

## 2 RELATED WORK

**Post-hoc Methods** Post-hoc calibration methods use hold-out validation data to learn calibration maps without modifying trained classifiers. Non-parametric approaches include Histogram Binning (HB) (Zadrozny & Elkan, 2001), its Bayesian extension BBQ (Naeini et al., 2015), and Spline

calibration (Gupta et al., 2021), though these often require more validation data and may alter prediction rankings. Parametric methods adjust outputs through predefined functional forms, including Temperature Scaling (TS) (Guo et al., 2017), enhanced variants PTS (Tomani et al., 2022) and CTS (Frenkel & Goldberger, 2021), Dirichlet Scaling (Kull et al., 2019) for multiclass calibration, Group Calibration (Yang et al., 2024), ProCal (Xiong et al., 2024) for proximity-based adjustments, and Feature Clipping (FC) (Tao et al., 2025). Ensemble-based post-hoc methods include data-augmentation ensembles (Conde et al., 2023) and Ensemble-based Temperature Scaling (ETS) (Zhang et al., 2020), though these demand significant computational resources. Conversely, our approach achieves superior calibration through more efficient means.

**Training Methods** Training-based calibration methods modify the learning process during model training to improve calibration, typically incurring higher computational costs. These include Brier Loss (Brier, 1950), MMCE (Kumar et al., 2018) with trainable calibration measures, Label Smoothing (Szegedy et al., 2016) that regularizes through softened target distributions, and Focal Loss variants (Mukhoti et al., 2020; Tao et al., 2023) addressing calibration through reweighting strategies. Ensemble-based training approaches include Deep ensembles (Lakshminarayanan et al., 2017) and dropout-based methods (Gal & Ghahramani, 2016) that leverage stochasticity as approximate Bayesian inference.

## 3 METHODOLOGY

We first present preliminaries in Section 3.1, then establish margin as a principled hardness indicator in Section 3.2. We identify fundamental limitations of NLL-based calibration objectives in Section 3.3, introduce our Charbonnier-SmoothSoftECE objective in Section 3.4, and present the SMART framework in Section 3.5.

### 3.1 PRELIMINARIES

A classification model is *calibrated* if its predictive confidence matches its actual accuracy. For classifier $f_\theta$, input $\mathbf{x}$ with true label $y$, and predicted class $\hat{y}$, perfect calibration requires $\mathbb{P}(y = \hat{y} \mid p_\theta(\hat{y} \mid \mathbf{x}) = p) = p$ for all confidence values $p \in [0, 1]$.

**Expected Calibration Error (ECE).** For classification model $f_\theta$ producing logits $\mathbf{z}_i \in \mathbb{R}^K$, the predictive probability for class $k$ is $p_\theta(y_i = k \mid \mathbf{x}_i) = \frac{\exp(z_{i,k})}{\sum_{j=1}^K \exp(z_{i,j})}$. To quantify calibration error, we partition samples into $B$ bins based on predicted confidence, compute average accuracy $\hat{a}_b$ and confidence $\hat{p}_b$ within each bin $b$, and measure their difference:

$$\text{ECE} = \sum_{b=1}^B \frac{|I_b|}{N} |\hat{p}_b - \hat{a}_b|, \tag{1}$$

where $I_b$ is the set of indices in bin $b$ and $N$ is the total number of samples.

**Smooth Calibration Error (smCE).** Beyond binned ECE which suffers from discretization artifacts, we also consider the smooth calibration error (Blasiok et al., 2023), defined as the worst-case correlation between the calibration residual and 1-Lipschitz probes of predicted confidence:

$$\text{smCE}(f) := \sup_{\varphi \in \mathcal{H}} \left| \mathbb{E}\left[ (a(X) - p(X))\varphi(p(X)) \right] \right|, \tag{2}$$

where $\mathcal{H} = \{\varphi : [0, 1] \to [-1, 1] \mid \text{Lip}(\varphi) \leq 1\}$ is the class of 1-Lipschitz continuous functions, $p(X)$ denotes the predicted confidence (maximum softmax probability), and $a(X) = \mathbb{I}\{\hat{y}(X) = y\}$ is the correctness indicator. This continuous metric avoids binning artifacts and provides theoretical foundation for our objective design in Section 3.4.

**Temperature Scaling.** Temperature scaling (TS) (Guo et al., 2017) introduces positive scalar $T$ to adjust logit distribution before softmax: $p_{\theta,T}(y_i = k \mid \mathbf{x}_i) = \frac{\exp(z_{i,k}/T)}{\sum_{j=1}^K \exp(z_{i,j}/T)}$. Smaller temperature $T < 1$ sharpens the distribution, while larger $T > 1$ flattens it. Vanilla TS finds global $\hat{T} = \arg\min_{T>0} \mathcal{L}_{\text{NLL}}(\mathcal{D}_{val}, f_\theta, T)$ by minimizing NLL on a validation set.

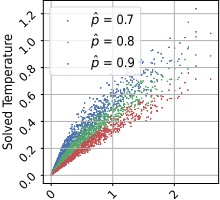 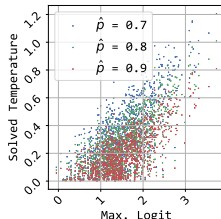 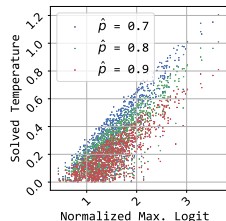 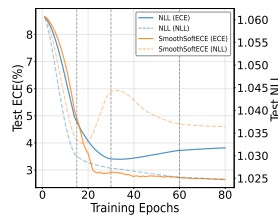

Figure 2: **Numerical study of temperature adjustment indicators.** The **left** three panels show joint distributions of solved temperature $T$ versus candidate indicators across 1,000 sampled logit vectors. **Right:** Test ECE (dashed) and NLL (solid) during SMART training on ImageNet ViT-B/32.

## 3.2 MARGIN AS A PRINCIPLED HARDNESS INDICATOR

Effective post-hoc calibration requires distinguishing between easy and hard samples to apply appropriate confidence adjustments. While existing methods (Xiong et al., 2024; Yang et al., 2024) recognize this need, they rely on indirectly learned proxies such as feature-space proximity or semantic clustering. We propose using the logit margin $m = z_{\max} - z_{2\text{nd}}$ as a direct hardness indicator, where $z_{\max}$ and $z_{2\text{nd}}$ are the largest and second-largest logits.

As demonstrated in Figure 1, samples with different margins exhibit systematically different calibration patterns even when sharing identical predicted confidence levels. Small-margin samples tend toward overconfidence while large-margin samples become underconfident, and margin correlates strongly with adversarial robustness ($r = 0.87$), confirming it captures proximity to decision boundaries. We now establish theoretically why margin provides superior temperature control compared to alternative indicators.

For a given logit vector $\mathbf{z} \in \mathbb{R}^K$ and target confidence $\hat{p} \in (0, 1)$, the temperature-confidence relationship $\frac{e^{z_{\max}/T}}{\sum_{k=1}^{K} e^{z_k/T}} = \hat{p}$ can be rearranged as $\sum_{k \neq M} e^{(z_k - z_{\max})/T} = S$ where $S := \frac{1}{\hat{p}} - 1$ and $M = \arg\max_k z_k$. Given only $z_{\max}$ and target $\hat{p} > 1/K$, we can construct configurations where all non-maximum logits equal $z_{\max} - \delta$ for varying $\delta > 0$, yielding $T = -\delta / \log(S/(K-1))$ which sweeps $(0, \infty)$ as $\delta$ varies. Thus maximum logit alone provides no bound on feasible temperatures.

In contrast, margin provides tight constraints. For any non-maximum class $k$, we have $z_k \leq z_{2\text{nd}} = z_{\max} - m$, leading to bounds $e^{-m/T} \leq S \leq (K-1)e^{-m/T}$. Solving for $T$ yields: when $\hat{p} > 1/2$, $T \in [\frac{m}{-\log(S/(K-1))}, \frac{m}{-\log S}]$ (finite interval); when $1/K < \hat{p} \leq 1/2$, $T \in [\frac{m}{-\log(S/(K-1))}, +\infty)$ (finite lower bound). The interval width decreases as $m$ grows, and for binary classification the bounds coincide to uniquely determine $T$. Complete derivations appear in Appendix A.1.

Figure 2 (left three panels) validates these results empirically. We sample 1,000 random logit vectors and numerically solve for temperatures achieving $\hat{p} = 0.8$. Margin exhibits clear functional structure with $T$ tightly constrained, while maximum logit and normalized maximum logit display scattered patterns spanning orders of magnitude. This establishes margin as the optimal scalar indicator for temperature-based calibration.

## 3.3 THE NLL-CALIBRATION MISMATCH

Current post-hoc calibration methods optimize negative log-likelihood (NLL) under the assumption that minimizing NLL improves calibration. We demonstrate this assumption can fail. Figure 2 (right-most panel) illustrates the phenomenon through a controlled experiment where we train SMART's margin-based temperature network on ImageNet ViT-B/32 using NLL as the training objective. While NLL decreases monotonically throughout 80 epochs, ECE begins increasing after epoch 30, creating clear divergence between objectives. By epoch 80, NLL has decreased by 15% while ECE has increased by 8% relative to epoch 30. This shows that following NLL gradients can actively worsen calibration despite improving likelihood.

We formalize conditions under which NLL and calibration objectives have opposing gradients. Consider a margin slice $G \subset [m_{\min}, m_{\max}]$ defining sample region $A := \{x : m(x) \in G\}$ where $m(x) = z_{(1)}(x) - z_{(2)}(x)$ is the margin between top two logits. We study local temperature scaling

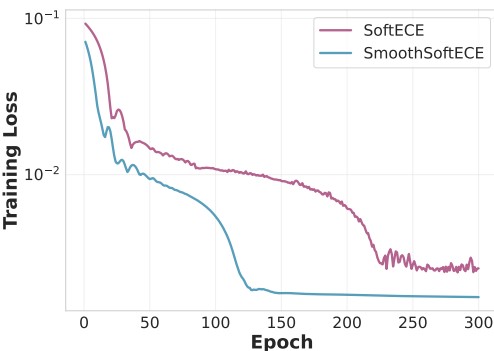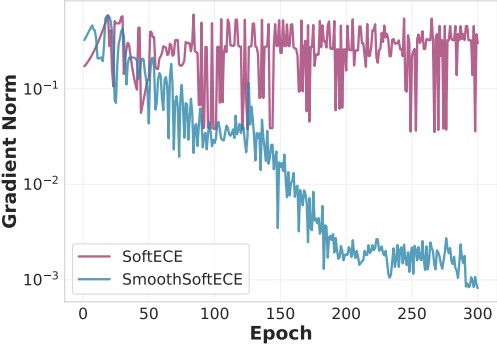

Figure 3: Post-hoc calibration on ImageNet ViT-B/16: training loss (left) and gradient norm (right) over training epochs SoftECE, and Charbonnier-SoftECE (ours).

by factor $s$ applied only to samples in $A$: $T_s(x) = T(x)/s$ if $x \in A$ and $T_s(x) = T(x)$ otherwise. At baseline $s = 1$, analyzing how objectives change as $s$ varies reveals their directional preferences.

Define $t_k := z_k/T$ (scaled logit), $q_k := \frac{e^{t_k}}{\sum_j e^{t_j}}$ (predicted probability), $\langle t \rangle_q := \sum_k q_k t_k$ (expected scaled logit), and $r_X(X) := \mathbb{P}(Y = M(X) \mid X)$ (pointwise top-class probability). For NLL $L_{\mathrm{nll}}(h_s)$ and calibration functional $\mathcal{C}[\psi] = \mathbb{E}[(a(X) - p_s)\psi(p_s)]$ with smooth probe $\psi$, the directional derivatives at $s = 1$ are:

$$\frac{d}{ds}L_{\mathrm{nll}}(h_s)\bigg|_{s=1} = \mathbb{E}\big[\mathbb{I}_A(t_Y - \langle t \rangle_q)\big], \tag{3}$$

$$\frac{d}{ds}\mathcal{C}[\psi]\bigg|_{s=1} = \mathbb{E}\big[\mathbb{I}_A\, p(t_M - \langle t \rangle_q)\big(\psi'(p)(r_X(X) - p) - \psi(p)\big)\big], \tag{4}$$

where $\mathbb{I}_A$ indicates the margin slice, $t_M$ is the scaled logit of the top class, and $t_Y$ is that of the true class. The NLL gradient depends only on whether $t_Y$ exceeds $\langle t \rangle_q$, while the calibration gradient depends on the calibration gap $r_X(X) - p$ weighted by probe sensitivity. These different sensitivities create potential for directional conflict.

Consider a margin slice $A$ with underconfident region $J_U$ having average calibration gap $\rho_U := \mathbb{E}[r_X(X) - p(X) \mid X \in J_U] > 0$ and overconfident region $J_O$ with gap $\rho_O := \mathbb{E}[p(X) - r_X(X) \mid X \in J_O] > 0$. Let $\mu_U, \mu_O$ denote relative proportions, $\gamma_{\min}, \gamma_{\max}$ be bounds on margin-to-temperature ratios in $A$, and $\Delta_G$ control logit spread. When underconfidence dominates NLL sensitivity such that $\rho_U \gamma_{\min} \mu_U > \gamma_{\max} \rho_O \mu_O + \Delta_G \mu_A$, there exists a sharpening direction where $\frac{d}{ds}L_{\mathrm{nll}}|_{s=1} < 0$ (NLL decreases) yet $\frac{d}{ds}\mathrm{smCE}|_{s=1} > 0$ (calibration worsens). The condition ensures that while sharpening helps underconfident samples in $J_U$, it harms overconfident samples in $J_O$ more severely, causing net calibration degradation despite NLL improvement. Detailed analysis appears in Appendix A.3.

This fundamental misalignment explains why NLL-based methods can achieve good likelihood while maintaining poor calibration. The mismatch occurs when calibration benefits from sharpening underconfident predictions are outweighed by costs from further sharpening overconfident predictions, yet NLL gradients favor overall sharpening due to different sensitivity to margin patterns. This motivates developing objectives that directly target calibration error rather than likelihood.

### 3.4 CHARBONNIER-SMOOTHED SOFTECE OBJECTIVE

Section 3.3 demonstrated that NLL optimization can conflict with calibration goals. We require a differentiable objective that directly targets calibration error while remaining statistically efficient with limited validation data. Current approaches face a bias-variance tradeoff: binned ECE has low variance but high bias from fixed binning, while point-wise losses have low bias but high variance from binary correctness indicators.

Following Karandikar et al. (2021), we adopt soft-binned ECE which balances this tradeoff through kernel smoothing. For sample $i$ with confidence $\hat{p}_i$ and bin centers $\{c_b\}_{b=1}^B$, soft membership weights

$w_{i,b} = \frac{\exp(-\alpha(\hat{p}_i - c_b)^2)}{\sum_{b'} \exp(-\alpha(\hat{p}_i - c_{b'})^2)}$ distribute each sample's contribution across neighboring bins, creating smooth gradients. In continuous formulation with Gaussian kernel $k_\lambda(t) = e^{-\lambda t^2}$ and reference density $\rho(u)$ on $[0, 1]$, this becomes:

$$\text{SoftECE}(f) := \mathbb{E}_X \left[ \int_0^1 K_\lambda(p(X), u)|a(X) - u|\rho(u)du \right], \tag{5}$$

where $K_\lambda(p, u) = \frac{k_\lambda(p-u)}{\int_0^1 k_\lambda(p-v)\rho(v)dv}$ is the normalized kernel, $p(X)$ is predicted confidence, and $a(X) = \mathbb{I}\{\hat{y}(X) = y\}$ is the correctness indicator.

We enhance SoftECE with Charbonnier smoothing to achieve theoretical control over calibration quality. Replacing the absolute value with Charbonnier function $\phi_\delta(r) = \sqrt{r^2 + \delta^2}$ yields:

$$\mathcal{H}_{\lambda,\delta}(f) := \mathbb{E}_X \left[ \int_0^1 K_\lambda(p(X), u)\phi_\delta(a(X) - u)\rho(u)du \right]. \tag{6}$$

The Charbonnier function provides $C^\infty$ smoothness while satisfying $\phi_\delta(r) \geq |r|$, ensuring that minimizing $\mathcal{H}_{\lambda,\delta}$ never weakens calibration control compared to the absolute value formulation. Our key theoretical contribution establishes that this objective provides an upper bound on smooth calibration error.

**Theorem 3.1** (Charbonnier-SoftECE Upper Bounds smCE). *Assume reference density $\rho$ satisfies $0 < \rho_{\min} \leq \rho(u) \leq \rho_{\max} < \infty$ for all $u \in [0, 1]$ with condition number $\kappa := \rho_{\max}/\rho_{\min}$. Then for all classifiers $f$ and smoothing parameters $\delta \geq 0$:*

$$\text{smCE}(f) \leq \mathcal{H}_{\lambda,\delta}(f) + 2B_\lambda, \tag{7}$$

*where $B_\lambda := \sup_{p \in [0,1]} \int_0^1 |p - u|K_\lambda(p, u)\rho(u)du$ represents kernel approximation error. For Gaussian kernels with $\lambda \geq 1$, $B_\lambda \leq \frac{C_\kappa}{\sqrt{\lambda}}$ where $C_\kappa := \frac{2\kappa}{\sqrt{\pi}\,\text{erf}(1)} \approx 1.339\kappa$.*

The proof (Appendix A.2) decomposes smCE using mollification: for any 1-Lipschitz probe $\varphi$, we write $\mathbb{E}[(a - p)\varphi(p)]$ as a smooth term controlled by $\mathcal{H}_{\lambda,\delta}$ plus approximation error bounded by $B_\lambda$. The bound splits into a model-dependent term $\mathcal{H}_{\lambda,\delta}(f)$ that can be optimized and a design-only term $2B_\lambda$ that tightens as $O(1/\sqrt{\lambda})$. Thus minimizing $\mathcal{H}_{\lambda,\delta}$ directly minimizes an upper bound on calibration error, resolving the NLL mismatch from Section 3.3.

Figure 3 demonstrates the practical benefits of Charbonnier smoothing. On ImageNet ViT-B/16, Charbonnier-SoftECE achieves faster training convergence (left panel) while maintaining stable gradient norms throughout optimization (right panel). Standard SoftECE exhibits oscillations in later training epochs. The Charbonnier enhancement thus provides both theoretical guarantees and improved optimization stability.

In practice, we discretize Equation equation 6 using $B = 15$ soft bins with Gaussian kernel bandwidth $\sigma = 0.05$ (corresponding to $\lambda = 200$) and Charbonnier parameter $\delta = 10^{-3}$. The bandwidth controls bias-variance tradeoff, the choice of hyperparameters $\lambda$ and $\delta$ exhibits stability across a reasonable range, as detailed in Appendix I.1.

## 3.5 THE SMART FRAMEWORK

Building on the theoretical foundations established in Sections 3.2–3.4, we introduce SMART (Sample Margin-Aware Recalibration of Temperature), which learns a direct mapping from margin to temperature. The framework combines margin as the input indicator (Section 3.2) with Charbonnier-SoftECE as the training objective (Section 3.4).

SMART implements a lightweight two-layer MLP that maps logit margin $m = z_{\max} - z_{\text{2nd}}$ to sample-specific temperature $T(m)$: $h = \text{ReLU}(W_1 m + b_1)$ and $T(m) = \text{softplus}(W_2 h + b_2) + \epsilon$, where the hidden dimension is 16 and $\epsilon = 10^{-1}$ ensures numerical stability. The softplus activation guarantees positive temperatures. This architecture requires only 49 trainable parameters regardless of the number of classes $K$, substantially fewer than existing parametric approaches: vector scaling requires $2K$ parameters, matrix scaling $K^2 + K$, class-dependent temperature scaling (CTS) (Frenkel & Goldberger, 2021) requires $K$, and spline calibration (Gupta et al., 2021) requires $13K$. For

ImageNet with $K = 1000$, these methods require thousands of parameters while SMART maintains minimal constant size.

Training minimizes the Charbonnier-SoftECE objective $\mathcal{H}_{\lambda,\delta}$ from Equation equation 6 on a validation set using Adam optimizer with initial learning rate $5 \times 10^{-3}$. For each sample, we compute its margin, predict temperature via the network, apply temperature scaling to logits, and compute the soft-binned calibration loss with Charbonnier smoothing. At inference, SMART computes the margin for each test sample, predicts its temperature through the trained network, and applies temperature scaling to obtain calibrated predictions. Complete training and inference procedures are detailed in Algorithm 1 (Appendix D).

## 4 EXPERIMENTS

### 4.1 EXPERIMENTAL SETUP

**Datasets** We conduct experiments on several benchmark datasets, including CIFAR-10, CIFAR-100 (Krizhevsky & Hinton, 2009), and ImageNet (Deng et al., 2009). To probe robustness under common corruptions and distribution shifts, we include *ImageNet-C* (All corruption type averaged, severity 5) (Hendrycks & Dietterich, 2019), *ImageNet-LT* (a long-tailed variant with power-law class imbalance) (Liu et al., 2019), and *ImageNet-Sketch* (sketch-based OOD variant) (Wang et al., 2019). All experiments employ a training-time batch size of 1024. CIFAR-10 and CIFAR-100 contain 60,000 images of size $32 \times 32$ pixels, with 10 and 100 classes, respectively, split into 45,000 training, 5,000 for validation and 10,000 test images. For ImageNet related dataset, we use 20% of the original test set, as the new validation set, with the remainder used as the test set. The testing batch size for all datasets is set to 128.

**Model Architectures.** To demonstrate the generality of our calibration methods, we evaluate across a diverse collection of convolutional and transformer–based networks. For CIFAR-10 and CIFAR-100, we employ ResNet-50 and ResNet-110 (He et al., 2016), Wide-ResNet (Zagoruyko & Komodakis, 2016), and DenseNet-121 (Huang et al., 2017), initialized with pretrained weights from Mukhoti *et al.* (Mukhoti et al., 2020). Each model is trained for 350 epochs using stochastic gradient descent with momentum 0.9, weight decay $5 \times 10^{-4}$, and a piecewise-constant learning-rate schedule $(0.1/0.01/0.001$ over $150/100/100$ epochs). ImageNet and its variants are evaluated on PyTorch's pretrained ResNet-50 and DenseNet-121 (Paszke et al., 2019), the transformer designs Swin-B (Liu et al., 2021), ViT-B/16 and ViT-B/32 (Dosovitskiy et al., 2021), and Wide-ResNet-50. This suite spans from compact CNNs to large-capacity transformers, allowing us to assess calibration robustness under varying architectural inductive biases and model complexities. Calibration performance is primarily evaluated using ECE, with additional metrics including AdaECE and top-1 accuracy. All experiments are conducted on a NVIDIA 3090 GPU, with results averaged over 5 seeds.

### 4.2 CALIBRATION PERFORMANCE

We evaluate SMART against leading post-hoc calibration approaches including TS (Guo et al., 2017), PTS (Tomani et al., 2022), CTS (Frenkel & Goldberger, 2021), and spline-based calibration (Gupta et al., 2021), Group Calibration (Yang et al., 2024), ProCal (Xiong et al., 2024), Feature Clipping (FC) (Tao et al., 2025), as well as uncalibrated (Vanilla) models across both standard settings and distribution shift scenarios.

**Calibration on Standard Datasets** SMART consistently outperforms these methods across CI-FAR10, CIFAR-100, and ImageNet-1K (Table 1), significantly reducing calibration error. The most notable improvement is seen in CIFAR-100, where SMART excels while Spline, despite its strong performance on other datasets, struggles. This highlights SMART's robustness across datasets with varying complexities. CNNs, which often suffer from overconfidence, are generally well-calibrated with TS-based methods. However, transformers see limited calibration improvements from TS-based methods, with SMART outperforming them by a large margin. On larger datasets like ImageNet-1K, SMART maintains its advantage with consistently lower ECE values. SMART works well on both CNN and ViTs where GC and FC failed.

Table 1: **Comparison of Post-Hoc Calibration Methods in ECE (%, ↓, 15 bins) Across Various Datasets and Models** (mean across 5 runs). The best-performing method for each dataset-model combination is in bold, and our method is highlighted. Full results with std are in App. E.

| Dataset | Model | Vanilla | TS 2017 | PTS 2022 | CTS 2021 | Spline 2021 | GC 2024 | ProCal 2024 | FC 2025 | SMART ours |
|---------|-------|---------|---------|----------|----------|-------------|---------|-------------|---------|------------|
| CIFAR-10 | ResNet-50 | 4.34 | 1.38 | 1.10 | 0.83 | 1.52 | 1.37 | 4.17 | 1.66 | **0.76** |
| | Wide-ResNet | 3.24 | 0.93 | 0.90 | 0.81 | 1.74 | 0.89 | 2.81 | 1.12 | **0.43** |
| CIFAR-100 | ResNet-50 | 17.53 | 5.61 | 1.96 | 3.67 | 3.48 | 5.70 | 9.71 | 2.91 | **1.37** |
| | Wide-ResNet | 15.34 | 4.50 | 1.96 | 3.01 | 3.76 | 4.55 | 9.44 | 4.49 | **1.80** |
| ImageNet-1K | ResNet-50 | 3.65 | 2.17 | 0.95 | 2.17 | 0.62 | 2.44 | 1.08 | 1.71 | **0.52** |
| | DenseNet-121 | 2.53 | 1.85 | 1.02 | 1.86 | 0.81 | 2.20 | 1.52 | 1.35 | **0.57** |
| | Wide-ResNet | 5.43 | 2.89 | 1.14 | 3.27 | 0.66 | 3.66 | 1.57 | 1.62 | **0.52** |
| | Swin-B | 5.05 | 3.91 | 1.05 | 1.53 | 0.88 | 4.95 | 1.00 | 5.05 | **0.46** |
| | ViT-B-16 | 5.62 | 3.60 | 1.23 | 4.65 | 0.91 | 4.39 | 0.97 | 5.65 | **0.48** |
| | ViT-B-32 | 6.39 | 3.93 | 1.27 | 2.12 | 0.81 | 4.67 | 0.88 | 6.39 | **0.71** |
| ImageNet-C | ResNet-50 | 13.82 | 1.97 | 1.12 | 1.69 | 5.61 | 2.69 | 5.79 | 2.51 | **0.62** |
| | DenseNet-121 | 12.57 | 1.58 | 1.19 | 1.44 | 5.18 | 2.01 | 9.88 | 9.44 | **0.63** |
| | Swin-B | 12.03 | 5.82 | 1.53 | 3.05 | 2.58 | 6.92 | 2.53 | 5.18 | **1.23** |
| | ViT-B-16 | 8.28 | 5.24 | 1.27 | 2.76 | 1.71 | 5.95 | 1.96 | 5.37 | **1.06** |
| | ViT-B-32 | 7.69 | 5.10 | 1.07 | 2.97 | 1.43 | 6.40 | 1.55 | 5.50 | **0.96** |
| ImageNet-LT | ResNet-50 | 3.63 | 2.01 | 0.99 | 2.17 | 0.56 | 2.20 | 1.12 | 1.80 | **0.56** |
| | DenseNet-121 | 2.50 | 1.80 | 1.20 | 1.88 | **0.79** | 2.05 | 1.79 | 1.76 | 0.81 |
| | Wide-ResNet | 5.40 | 2.99 | 1.21 | 2.87 | 0.81 | 3.59 | 1.28 | 1.68 | **0.53** |
| | Swin-B | 4.69 | 3.98 | 1.21 | 1.50 | 0.79 | 4.79 | 0.95 | 4.82 | **0.58** |
| | ViT-B-16 | 5.58 | 3.73 | 1.14 | 1.43 | 0.66 | 4.34 | 0.77 | 5.72 | **0.56** |
| | ViT-B-32 | 6.28 | 3.98 | 1.35 | 2.12 | 0.72 | 4.76 | 0.83 | 6.26 | **0.60** |
| ImageNet-S | ResNet-50 | 22.32 | 2.06 | 1.69 | 1.48 | 9.76 | 1.99 | 9.52 | 12.58 | **0.92** |
| | DenseNet-121 | 20.13 | 1.67 | 1.93 | 1.16 | 9.20 | 1.77 | 12.93 | 22.67 | **0.59** |
| | Swin-B | 24.61 | 6.50 | 1.53 | 3.62 | 8.66 | 6.92 | 8.05 | 1.70 | **1.26** |
| | ViT-B-16 | 16.57 | 5.75 | 1.33 | 2.84 | 5.70 | 6.36 | 5.67 | 1.93 | **0.98** |
| | ViT-B-32 | 14.22 | 4.99 | 1.67 | 3.25 | 4.07 | 6.23 | 4.44 | 1.56 | **0.87** |

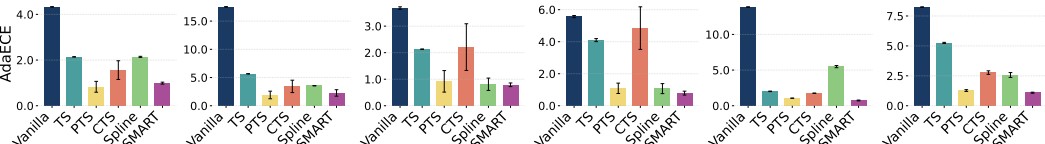

Figure 4: **Comparison of calibration methods using AdaECE↓ across various datasets and models.** From left to right: CIFAR-10 (ResNet-50), CIFAR-100 (ResNet-50), ImageNet (ResNet-50), ImageNet (ViT-B-16), ImageNet-C (ResNet-50), and ImageNet-C (ViT-B-16). Results are averaged.

**Robustness under Class Imbalance and Distribution Shift** Across long-tailed (ImageNet-LT) and corrupted scenarios (ImageNet-Sketch, ImageNet-C), SMART's sample-wise temperature adaptation consistently outperforms global and class-wise scalers. Uniform approaches such as TS struggle to accommodate underrepresented classes or severe input degradations, leading to pronounced calibration drift. Spline, FC and ProCal failed on Imagenet-S with CNNs where SMART still performs robustly.

**Calibration Performance on AdaECE** We also evaluate SMART using Adaptive Expected Calibration Error (AdaECE) to provide a comprehensive view of its performance, shown in Figure 4, with additional results available in Appendix F. SMART demonstrates superior performance on AdaECE compared to traditional calibration methods across diverse settings. AdaECE addresses limitations of standard ECE by accounting for uneven confidence distributions, providing a more reliable measure of calibration quality. SMART consistently achieves the lowest AdaECE values and variance across CNN and ViT architectures and datasets (CIFAR and ImageNet variants), demonstrating its robustness

to dataset shifts and model architectures. Notably, SMART outperforms more complex methods like Spline calibration and CTS in calibration error and variance while requiring fewer parameters.

By leveraging instance-level temperature through logit margins, SMART yields stable calibration gains across diverse distribution shifts. Its lightweight per-sample inference preserves efficiency while delivering robustness that neither fixed nor ensemble temperature schemes can match. In contrast, Spline collapses on particularly challenging shifts such as ImageNet-S and ImageNet-C — whereas our method consistently sustains the lowest and most stable calibration error even under these adverse conditions.

### 4.3 COMPARISON WITH TRAINING-TIME CALIBRATION METHODS

We evaluate SMART alongside training-time calibration techniques in Table 2, including Brier Loss (Brier, 1950), Maximum Mean Calibration Error (MMCE) (Kumar et al., 2018), Label Smoothing (LS-0.05) (Szegedy et al., 2016), and Focal Loss variants (FLSD-53 and FL-3) (Mukhoti et al., 2020). This shows that combining SMART with these methods consistently enhances calibration performance across various models and datasets, further validating SMART's effectiveness alongside training-time approaches. Moreover, as seen in Table 1, SMART alone, as a post-hoc calibration method, already outperforms these train-time techniques with minimal computational overhead, while train-time methods require significantly more resources.

Table 2: **Comparison of Train-time Calibration Methods Using ECE($\downarrow$, %, 15 bins) Across Various Datasets and Models.** The best-performing method for each dataset-model combination is in bold, and our method (SMART) is highlighted. Results are averaged over 5 runs.

| Dataset | Model | NLL | | Brier Loss | | MMCE | | LS-0.05 | | FLSD-53 | | FL-3 | |
|---|---|---|---|---|---|---|---|---|---|---|---|---|---|
| | | base | ours | base | ours | base | ours | base | ours | base | ours | base | ours |
| CIFAR10 | ResNet-50 | 4.34 | **0.75** | 1.81 | **0.96** | 4.57 | **0.53** | 2.97 | **0.51** | 1.56 | **0.42** | 1.47 | **0.43** |
| | ResNet-110 | 4.41 | **0.44** | 2.56 | **0.60** | 5.07 | **0.38** | 2.09 | **0.28** | 1.87 | **0.45** | 1.54 | **0.54** |
| | DenseNet-121 | 4.51 | **0.53** | 1.52 | **0.31** | 5.10 | **0.66** | 1.89 | **0.51** | 1.23 | **0.62** | 1.31 | **1.02** |
| | Wide-ResNet | 3.24 | **0.30** | 1.25 | **0.38** | 3.30 | **0.34** | 4.25 | **0.36** | 1.58 | **0.39** | 1.68 | **0.54** |
| CIFAR100 | ResNet-50 | 17.53 | **0.99** | 6.54 | **1.01** | 15.31 | **0.86** | 7.81 | **1.50** | 4.49 | **1.26** | 5.16 | **0.56** |
| | ResNet-110 | 19.06 | **0.98** | 7.87 | **0.87** | 19.13 | **1.42** | 11.03 | **1.01** | 8.54 | **0.85** | 8.65 | **0.73** |
| | DenseNet-121 | 20.99 | **1.86** | 5.22 | **0.59** | 19.10 | **1.34** | 12.87 | **1.02** | 3.70 | **0.91** | 4.14 | **0.98** |
| | Wide-ResNet | 15.34 | **1.38** | 4.35 | **1.00** | 13.17 | **0.98** | 4.88 | **1.24** | 3.02 | **0.79** | 2.14 | **1.12** |

### 4.4 SCALABILITY WITH VALIDATION DATA

**Scalability with Validation Data** SMART demonstrates superior ability to leverage increasing validation sample sizes compared to competing calibration methods, shown in Figure 5. While all approaches struggle with minimal validation data, SMART exhibits continuous performance improvement throughout the entire range of sample sizes tested, ultimately achieving the lowest calibration error. In contrast, the alternative methods display more limited utilization of additional validation samples. TS reaches a performance plateau at moderate sample sizes and fails to improve further, while PTS exhibits concerning instability in the mid-range sample sizes, implicitly reflecting the NLL mismatch. GC demonstrates the most problematic behavior, with significant performance spikes that indicate poor robustness to varying validation set sizes. The consistent improvement trajectory of SMART manifests the margin provides a robust signal that enables more

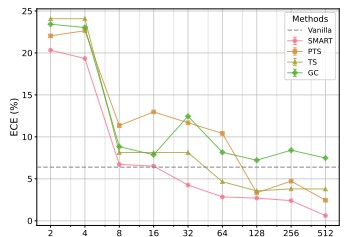

Figure 5: **ECE($\downarrow$, %, 15 bins) versus validation sample size.** Comparison of calibration methods on ImageNet (ViT-B/32), averaged over five runs.

effective temperature estimation as additional validation samples become available. This superior sample utilization capability makes SMART particularly valuable in practical applications where validation data availability may vary.

## 4.5 ABLATION STUDIES

**Choice of Calibration Objective**   We compare how various calibration objective influence SMART's calibration performance (Table 3). While all tested objectives enable significant improvements over vanilla, they exhibit distinct behavior patterns across architectures. NLL and label smoothing losses, despite their prevalence in classification tasks, demonstrate suboptimal calibration performance due to their indirect relationship with confidence estimation objectives. MSE and Brier score offer more reliable improvements by directly penalizing squared confidence errors, yet their effectiveness fluctuates between CNN and transformer architectures. Charbonnier-SoftECE emerges as the superior choice by directly optimizing the calibration metric itself, achieving both the lowest average error and the smallest variance across diverse model architectures, making it the most stable choice for SMART's temperature mapping.

Table 3: **Different Calibration Objective.** ECE (%, ↓, 15 bins) on ImageNet averaged over 5 runs.

| Architecture | Method | NLL | LS | MSE | Brier | SoftECE | Charbonnier-SoftECE |
|---|---|---|---|---|---|---|---|
| ResNet-50 (Top-1 = 0.761) | TS | 2.04 | 14.33 | 3.69 | 2.31 | 3.16 | 2.12 |
| | PTS | 1.04 | 1.87 | 1.89 | 1.88 | 1.88 | 0.94 |
| | **SMART** | **0.93** | **1.09** | **1.39** | **1.38** | **0.65** | **0.52** |
| ViT-B/16 (Top-1 = 0.810) | TS | 3.73 | 6.05 | 5.58 | 3.11 | 3.10 | 3.08 |
| | PTS | 5.69 | 3.22 | 2.40 | 2.57 | 1.15 | 0.77 |
| | **SMART** | **3.62** | **3.11** | **0.84** | **0.80** | **0.89** | **0.48** |

Table 4: **Comparison on alternative on-the-shelf indicator** on ImageNet-1K.

| Model | Entropy | Conf. | All Logits | $Logit_{max}$ | $Logit_{max}$ - $\overline{Logits}$ | Margin (ours) |
|---|---|---|---|---|---|---|
| ResNet-50 | 0.87 | 0.97 | 0.87 | 0.91 | 0.85 | **0.58** |
| DenseNet-121 | 0.62 | 0.89 | 0.79 | 0.80 | 0.84 | **0.56** |
| Wide-ResNet | 1.00 | 1.22 | 0.92 | 0.57 | 0.63 | **0.52** |
| Swin-B | 0.62 | 0.81 | 0.89 | 0.78 | 0.87 | **0.63** |
| ViT-B/16 | 0.90 | 0.75 | 0.97 | 0.91 | 1.20 | **0.72** |

**Choice of Indicators**   We evaluated six candidate uncertainty signals as inputs to our temperature network on ImageNet-1K (Table 4): predictive entropy, predicted confidence, full logit vectors, maximum logit, mean-normalized logit deviation, and our proposed margin. The margin consistently achieves the lowest calibration error across all tested architectures, outperforming alternative indicators by substantial margins. While full logit vectors contain rich information, they introduce excessive noise that degrades performance in limited-data scenarios. Simpler scalar measures like maximum logit or predicted confidence fail to adequately capture the competitive dynamics between top classes that drive miscalibration. The margin's superior performance stems from its ability to distill prediction uncertainty into a minimal yet complete representation that directly reflects decision boundary proximity, enabling robust calibration across diverse model architectures.

## 5 CONCLUSION AND LIMITATION

We introduced SMART, a lightweight recalibration method leveraging the logit margin as a principled calibration indicator for precise temp adjustment. By capturing sample hardness through this indicative signal, SMART achieves SOTA calibration performance with minimal parameters compared to existing methods. Our Charbonnier-Smoothed SoftECE objective enables stable optimization as validation data scales. Extensive experiments confirm SMART's robustness across diverse architectures, datasets, and challenging distribution shifts, consistently outperforming current post-hoc and even training-based methods. Future work could explore integrating SMART with other uncertainty quantification methods or investigate other hardness indicator to further improve calibration and robustness in safety-critical applications.

**Limitation**   While SMART demonstrates excellent performance across tested scenarios, its effectiveness may vary slightly for extremely specialized domains with highly skewed class distributions. Additionally, though our method requires minimal validation data, performance could degrade in zero-shot scenarios where no domain-specific calibration samples are available.

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

## A  THEORETICAL PROOFS

### A.1  TEMPERATURE–CONFIDENCE RELATION AND MARGIN BOUNDS

**Problem and observation.** To reach a target top-class confidence $\hat{p} \in (0, 1)$, how constrained is $T$? Empirically, fixing only $z_M$ leaves $T$ ill-determined; using the *Margin* $m$ yields tight bounds. We now prove this, step by step.

**Target confidence equation.**  Requiring $p_{\phi,\max} = \hat{p}$ is equivalent to

$$\frac{e^{z_M/T}}{\sum_{j=1}^{K} e^{z_j/T}} = \hat{p} \qquad \Longleftrightarrow \qquad \sum_{j \neq M} e^{(z_j - z_M)/T} = \frac{1}{\hat{p}} - 1 \; := \; S. \tag{8}$$

Because $\hat{p}$ is the *maximum* softmax probability, $\hat{p} \geq 1/K$, hence $S \leq K-1$; moreover $S > 0$ since $\hat{p} < 1$. Thus $S \in (0, K-1]$, and if we assume a strict top-1 margin $m > 0$ (no top-2 ties) then $\hat{p} > 1/K$ and $S \in (0, K-1)$.

**Unboundedness if only $z_M$ is known.**  Assume $z_j - z_M = -\delta$ for all $j \neq M$ with $\delta > 0$. Then

$$\sum_{j \neq M} e^{(z_j - z_M)/T} = \sum_{j \neq M} e^{-\delta/T} = (K-1)\, e^{-\delta/T} = S, \tag{9}$$

$$\Rightarrow \quad T = -\frac{\delta}{\log\left(\frac{S}{K-1}\right)}. \tag{10}$$

When $S < K-1$ (equivalently $\hat{p} > 1/K$), $\log(S/(K-1)) < 0$, so as $\delta \in (0, \infty)$ varies, equation 10 sweeps $T \in (0, \infty)$. Thus

> Fixing $z_M$ alone leaves the feasible $T$ unbounded: $(0, \infty)$.

**$m$-boundedness.**  Let $\mu \in \arg\max_{j \neq M} z_j$ denote a runner-up index and define the (nonnegative) margin $m := z_M - z_\mu$. For any $j \notin \{M, \mu\}$,

$$z_j - z_M \; \leq \; z_\mu - z_M \; = \; -m \quad \Longrightarrow \quad e^{(z_j - z_M)/T} \; \leq \; e^{-m/T}.$$

Since $e^{(z_\mu - z_M)/T} = e^{-m/T}$, we have

$$\sum_{j \neq M} e^{(z_j - z_M)/T} = e^{-m/T} + \sum_{j \notin \{M,\mu\}} e^{(z_j - z_M)/T}$$

$$\geq e^{-m/T}, \tag{11}$$

$$\sum_{j \neq M} e^{(z_j - z_M)/T} \leq e^{-m/T} + (K-2)\, e^{-m/T} = (K-1)\, e^{-m/T}. \tag{12}$$

Combining equation 8, equation 11, and equation 12 yields

$$e^{-m/T} \; \leq \; S \; \leq \; (K-1)\, e^{-m/T}, \qquad S = \tfrac{1}{\hat{p}} - 1 \in (0, K-1]. \tag{13}$$

Equivalently,

$$-\log\left(\tfrac{S}{K-1}\right) \; \geq \; \frac{m}{T} \; \geq \; -\log S. \tag{14}$$

Solving equation 14 for $T > 0$ gives the $m$-bounded feasible set:

$$\begin{cases} T \in \left[ \dfrac{m}{-\log(S/(K-1))}, \; \dfrac{m}{-\log S} \right], & \text{if } 0 < S < 1 \; (\hat{p} > 1/2), \\[2ex] T \in \left[ \dfrac{m}{-\log(S/(K-1))}, \; \infty \right), & \text{if } 1 \leq S < K-1 \; (1/K < \hat{p} \leq 1/2), \\[2ex] \text{feasible iff } m = 0, & \text{if } S = K-1 \; (\hat{p} = 1/K). \end{cases}$$

*Interpretation.* Knowing the Margin $m$ pins down $T$ tightly. When the target confidence is above $1/2$, the feasible $T$ is a *finite* interval whose width shrinks as $m$ grows. For lower target confidences ($\hat{p} \leq 1/2$), one still gets a nontrivial *lower* bound on $T$; a finite upper bound appears exactly when $S < 1$ (i.e., $\hat{p} > 1/2$). In particular, for $K = 2$ the bounds coincide and $T = \frac{m}{-\log S}$ is uniquely determined.

## A.2 CHARBONNIER–SOFTECE UPPER-BOUNDS SMCE

**Setup and goal.** Let $p(x) \in [0, 1]$ denote the predicted probability of correctness (top-class confidence) and $a(x) := \mathbb{1}\{\hat{y}(x) = y\} \in \{0, 1\}$ the correctness indicator. We measure calibration via the *smooth calibration error* (smCE), the worst-case correlation between the residual $a(X) - p(X)$ and any 1-Lipschitz probe of the prediction $p(X)$ (cf. forecasting Kakade & Foster (2008)) and the ML calibration view of Blasiok et al. (2023)):

$$\text{smCE}(f) := \sup_{\varphi \in \mathcal{H}} \Big| \mathbb{E}\big[(a(X) - p(X)) \varphi\big(p(X)\big)\big] \Big|,$$

$$\mathcal{H} := \big\{ \varphi : [0, 1] \to [-1, 1] \text{ s.t. } \text{Lip}(\varphi) \leq 1 \big\}.$$

We study the *Charbonnier–SoftECE* objective (a smoothed, Huberized absolute calibration error):

$$\mathcal{H}_{\lambda,\delta}(f) := \mathbb{E}_X\left[\int_0^1 K_\lambda\big(p(X), u\big) \phi_\delta\big(a(X) - u\big) \rho(u)\, du\right], \qquad \phi_\delta(r) := \sqrt{r^2 + \delta^2},$$

where $\rho$ is a reference density on $[0, 1]$ and

$$K_\lambda(p, u) = \frac{k_\lambda(p - u)}{\int_0^1 k_\lambda(p - v)\, \rho(v)\, dv}, \qquad k_\lambda(t) := e^{-\lambda t^2}, \; \lambda > 0, \tag{15}$$

so that $\int_0^1 K_\lambda(p, u)\, \rho(u)\, du = 1$ for every $p \in [0, 1]$. Assume *boundedness and bounded-away-from-zero* of $\rho$: there exist constants $0 < \rho_{\min} \leq \rho(u) \leq \rho_{\max} < \infty$ for all $u \in [0, 1]$, and write $\kappa := \rho_{\max}/\rho_{\min}$.

**Main result.**

**Theorem A.1** (Charbonnier–SoftECE upper-bounds smCE). *Under the assumptions above, for all classifiers $f$ and all $\delta \geq 0$,*

$$\text{smCE}(f) \leq \mathcal{H}_{\lambda,\delta}(f) + 2 B_\lambda, \qquad B_\lambda := \sup_{p \in [0,1]} \int_0^1 |p - u|\, K_\lambda(p, u)\, \rho(u)\, du. \tag{16}$$

*Moreover, for the Gaussian kernel $k_\lambda(t) = e^{-\lambda t^2}$,*

$$B_\lambda \leq \min\left\{ 1, \frac{2\kappa}{\sqrt{\pi}} \cdot \frac{1}{\sqrt{\lambda}\, \text{erf}(\sqrt{\lambda})} \right\}, \tag{17}$$

*and in particular for $\lambda \geq 1$,*

$$B_\lambda \leq \frac{C_\kappa}{\sqrt{\lambda}}, \qquad C_\kappa := \frac{2\kappa}{\sqrt{\pi}\, \text{erf}(1)} \approx 1.339\, \kappa. \tag{18}$$

**Proof.** For brevity write $p := p(X)$ and $a := a(X)$. Fix any $\varphi \in \mathcal{H}$ with $\|\varphi\|_\infty \leq 1$. Introduce the (normalized) kernel smoothing operator

$$(\mathsf{T}_\lambda \varphi)(p) := \int_0^1 K_\lambda(p, u)\, \varphi(u)\, \rho(u)\, du.$$

*Decomposition.*

$$\mathbb{E}[(a - p) \varphi(p)] = \mathbb{E}[(a - p) (\mathsf{T}_\lambda \varphi)(p)] + \mathbb{E}[(a - p) \{\varphi(p) - (\mathsf{T}_\lambda \varphi)(p)\}]. \tag{19}$$

*Approximation (mollification) error.* By $\text{Lip}(\varphi) \leq 1$ and the triangle inequality,

$$\big|\varphi(p) - (\mathsf{T}_\lambda \varphi)(p)\big| = \left|\int_0^1 K_\lambda(p, u) \{\varphi(p) - \varphi(u)\}\, \rho(u)\, du\right| \leq \int_0^1 K_\lambda(p, u)\, |p - u|\, \rho(u)\, du.$$

Taking the supremum over $p$ and $\varphi$ yields

$$\sup_{\varphi \in \mathcal{H}} \sup_{p \in [0,1]} \big|\varphi(p) - (\mathsf{T}_\lambda \varphi)(p)\big| \leq B_\lambda. \tag{20}$$

Hence, using $|a - p| \leq 1$,

$$\left| \mathbb{E}\big[(a - p)\left\{\varphi(p) - (\mathsf{T}_\lambda \varphi)(p)\right\}\big] \right| \; \leq \; B_\lambda.$$

*Aligned main term.* By Fubini/Tonelli (bounded integrands) and normalization $\int K_\lambda(p, u)\rho(u)\,du = 1$,

$$\mathbb{E}[(a - p)\,(\mathsf{T}_\lambda \varphi)(p)] = \int_0^1 \varphi(u)\,\mathbb{E}\big[(a - p)\,K_\lambda(p, u)\big]\,\rho(u)\,du,$$

and since $|\varphi(u)| \leq 1$,

$$\left| \mathbb{E}[(a - p)\,(\mathsf{T}_\lambda \varphi)(p)] \right| \; \leq \; \int_0^1 \left| \mathbb{E}\big[(a - p)\,K_\lambda(p, u)\big] \right|\rho(u)\,du. \tag{21}$$

For each $u$, using $|x + y| \leq |x| + |y|$ and $|a - p| \leq |a - u| + |p - u|$, together with $\phi_\delta(r) \geq |r|$,

$$\left| \mathbb{E}\big[(a - p)\,K_\lambda(p, u)\big] \right| \leq \mathbb{E}\big[|a - p|\,K_\lambda(p, u)\big]$$
$$\leq \; \mathbb{E}\big[\phi_\delta(a - u)\,K_\lambda(p, u)\big] \; + \; \mathbb{E}\big[|p - u|\,K_\lambda(p, u)\big]. \tag{22}$$

Integrating equation 22 against $\rho(u)\,du$ and applying Fubini,

$$\int_0^1 \left| \mathbb{E}\big[(a - p)\,K_\lambda(p, u)\big] \right|\rho(u)\,du \; \leq \; \mathbb{E}\left[\int_0^1 K_\lambda(p, u)\,\phi_\delta(a - u)\,\rho(u)\,du\right]$$
$$+ \; \sup_p \int_0^1 |p - u|\,K_\lambda(p, u)\,\rho(u)\,du,$$

i.e.,

$$\left| \mathbb{E}[(a - p)\,(\mathsf{T}_\lambda \varphi)(p)] \right| \; \leq \; \mathcal{H}_{\lambda,\delta}(f) \; + \; B_\lambda. \tag{23}$$

*Conclusion.* Combining equation 20, equation 23 with equation 19 and taking the supremum over $\varphi \in \mathcal{H}$ gives $\mathrm{smCE}(f) \leq \mathcal{H}_{\lambda,\delta}(f) + 2B_\lambda$.

*Explicit bounds for $B_\lambda$.* By definition,

$$B_\lambda = \sup_{p \in [0,1]} \frac{\int_0^1 |p - u|\,k_\lambda(p - u)\,\rho(u)\,du}{\int_0^1 k_\lambda(p - v)\,\rho(v)\,dv}.$$

Using $\rho(u) \leq \rho_{\max}$ in the numerator and $\rho(v) \geq \rho_{\min}$ in the denominator, and changing variables $t = p - u$ or $t = p - v$, we obtain for all $p \in [0, 1]$:

$$B_\lambda \; \leq \; \kappa \cdot \frac{\int_\mathbb{R} |t|\,e^{-\lambda t^2}\,dt}{\int_{p-1}^p e^{-\lambda t^2}\,dt}.$$

Since $p \in [0, 1]$, the denominator integrates over a length-1 interval contained in $[-1, 1]$; by symmetry and unimodality of $t \mapsto e^{-\lambda t^2}$, the minimum over such intervals is attained at an endpoint, e.g. $[0, 1]$. Hence

$$B_\lambda \; \leq \; \kappa \cdot \frac{\int_\mathbb{R} |t|\,e^{-\lambda t^2}\,dt}{\int_0^1 e^{-\lambda t^2}\,dt} \; = \; \kappa \cdot \frac{\frac{1}{\lambda}}{\frac{\sqrt{\pi}}{2\sqrt{\lambda}}\,\mathrm{erf}(\sqrt{\lambda})} \; = \; \frac{2\,\kappa}{\sqrt{\pi}} \cdot \frac{1}{\sqrt{\lambda}\,\mathrm{erf}(\sqrt{\lambda})}.$$

Since $|p - u| \leq 1$ and $\int K_\lambda \rho = 1$, we also have $B_\lambda \leq 1$. For $\lambda \geq 1$, $\mathrm{erf}(\sqrt{\lambda}) \geq \mathrm{erf}(1)$, yielding equation 18. $\qquad\square$

**Interpretation and guidance.** The guarantee equation 16 decomposes into a *model-dependent* term $\mathcal{H}_{\lambda,\delta}(f)$ and a *design-only* kernel bias $B_\lambda$, the average *soft-bin radius* around $p$. The Charbonnier envelope obeys $\phi_\delta(r) \geq |r|$, so replacing $|a - u|$ with $\phi_\delta(a - u)$ never weakens control of smCE and yields smooth gradients near $r = 0$. For Gaussian kernels, $B_\lambda = \mathcal{O}(\kappa/\sqrt{\lambda})$ as in equation 18, so increasing $\lambda$ monotonically tightens the bound; the cap $B_\lambda \leq 1$ ensures uniform validity for all $\lambda > 0$. (Discrete soft-binned implementations—via Riemann-sum quadrature of the $u$-integral—inherit the same inequality up to a standard design-only quadrature error that vanishes as the grid is refined.)

## A.3 CHARBONNIER–SOFTECE VS. NLL

We compare negative log-likelihood (NLL) with Charbonnier–SoftECE within the SMART family $T(x) = h(m(x))$ that scales by the *margin* $m(x) := z_{(1)}(x) - z_{(2)}(x) \in \mathbb{R}_{\geq 0}$. Throughout, assume (i) $T(x) \in [T_{\min}, T_{\max}]$ with $0 < T_{\min} \leq T_{\max} < \infty$, (ii) $\mathbb{E}\|z(X)\|_\infty < \infty$, and (iii) $\mathbb{P}(z_{(1)} = z_{(2)}) = 0$ (no top-2 ties a.s.). Write $t(x) := z(x)/T(x)$, $q(x) := \mathrm{softmax}(t(x))$, $M(x) := \arg\max_k t_k(x)$, $p(x) := q_{M(x)}(x) \in (0,1)$, and $Y^\top(x) := \mathbf{1}\{Y(x) = M(x)\}$. Define the pointwise top-class probability $r_X(x) := \mathbb{P}(Y = M(x) \mid X = x)$ and the *reliability* curve $r(p) := \mathbb{E}[r_X(X) \mid p(X) = p]$. We measure calibration by the smooth calibration error (smCE)

$$\mathrm{smCE}(f) := \sup_{\substack{\varphi:[0,1]\to[-1,1] \\ \mathrm{Lip}(\varphi)\leq 1}} \left| \mathbb{E}[(Y^\top - p)\varphi(p)] \right|.$$

**Charbonnier–SoftECE and its smCE control.** Charbonnier–SoftECE is the objective

$$\mathcal{H}_{\lambda,\delta}(f) := \mathbb{E}_X \left[ \int_0^1 K_\lambda(p(X), u)\, \phi_\delta(Y^\top(X) - u)\, \rho(u)\, du \right], \qquad \phi_\delta(r) := \sqrt{r^2 + \delta^2},$$

with normalized kernel $K_\lambda$ as in equation 15 and a reference density $\rho$ on $[0,1]$. We *use* (proved in Sec. A.2) the smCE control

$$\mathrm{smCE}(f) \leq \mathcal{H}_{\lambda,\delta}(f) + 2\, B_\lambda, \qquad B_\lambda = \sup_p \int_0^1 |p - u|\, K_\lambda(p, u)\, \rho(u)\, du, \qquad (24)$$

with $B_\lambda = \mathcal{O}(\kappa/\sqrt{\lambda})$ for Gaussian kernels.

**A SMART-feasible local scaling path.** Fix a Borel margin slice $G \subset \mathbb{R}_{\geq 0}$ and $A := \{x : m(x) \in G\}$. For $s > 0$, define the local scaling

$$T_s(x) := \begin{cases} T(x)/s, & x \in A, \\ T(x), & x \notin A, \end{cases}$$

$$t_s(x) := \frac{z(x)}{T_s(x)} = \begin{cases} s\, t(x), & x \in A, \\ t(x), & x \notin A, \end{cases}$$

$$q^{(s)} := \mathrm{softmax}(t_s), \quad p_s := q_M^{(s)}.$$

Because uniform multiplication by $s > 0$ preserves coordinate ordering, $M$ is unchanged for all $s > 0$; (iii) rules out measure-zero ties at the boundary.

**Lemma 1** (Directional derivatives under local margin-dependent scaling)**.** *Let* $L_{\mathrm{nll}}(h) := \mathbb{E}[-\log q_Y(X)]$. *For any $C^1$ probe $\psi : [0,1] \to \mathbb{R}$ with $\mathrm{Lip}(\psi) \leq 1$ and $\|\psi\|_\infty \leq 1$, the Gâteaux derivatives at $s = 1$ exist and*

$$\left. \frac{d}{ds} L_{\mathrm{nll}}(h_s) \right|_{s=1} = \mathbb{E}[\mathbf{1}_A (\langle t \rangle_q - t_Y)], \qquad (25)$$

$$\left. \frac{d}{ds} \mathbb{E}[(Y^\top - p_s)\psi(p_s)] \right|_{s=1} = \mathbb{E}\Big[\mathbf{1}_A\, p\, (t_M - \langle t \rangle_q)\, (\psi'(p)\, (r_X - p) - \psi(p))\Big], \qquad (26)$$

*where $\langle t \rangle_q := \sum_k q_k t_k$ and $r_X := r_X(X)$.*

*Proof.* On $A$, $\partial_s q_k^{(s)} = q_k^{(s)}(t_k - \langle t \rangle_{q^{(s)}})$, hence $\partial_s(-\log q_Y^{(s)}) = \langle t \rangle_{q^{(s)}} - t_Y$. Outside $A$ the derivative vanishes. Dominated convergence applies since $|\partial_s(-\log q_Y^{(s)})| \leq 2\|t\|_\infty$ and $\mathbb{E}\|t\|_\infty \leq \mathbb{E}\|z\|_\infty/T_{\min} < \infty$, yielding equation 25. For $F_\psi(s) := \mathbb{E}[(Y^\top - p_s)\psi(p_s)]$, with $M$ fixed, $\partial_s p_s = \partial_s q_M^{(s)} = q_M^{(s)}(t_M - \langle t \rangle_{q^{(s)}}) = p_s(t_M - \langle t \rangle_{q^{(s)}})$. Thus

$$\partial_s\big((Y^\top - p_s)\psi(p_s)\big) = \big(-\psi(p_s) + (Y^\top - p_s)\psi'(p_s)\big)\, \partial_s p_s.$$

Conditioning on $X$ replaces $Y^\top$ by $r_X(X)$, whence equation 26 at $s = 1$ after integration; dominated convergence holds because $p\, |t_M - \langle t \rangle_q| \leq 2\|t\|_\infty$ and $|\psi'| \leq 1$, $|\psi| \leq 1$. $\qquad\square$

**Lemma 2** (Margin lower bound for the top-logit advantage). *On* $\{M = \arg\max t\}$,

$$t_M - \langle t \rangle_q \ \geq \ (1-p)\left(t_M - t_{(2)}\right) \ = \ (1-p)\,\frac{m}{T}. \tag{27}$$

*Proof.* $\langle t \rangle_q = p\, t_M + \sum_{j \neq M} q_j t_j \leq p\, t_M + (1-p)\, t_{(2)}$; rearrange. $\square$

**A correct NLL directional upper bound (multi-class).** Define the *runner-up gap* $g(x) := t_{(1)}(x) - t_{(2)}(x) = m(x)/T(x) \geq 0$ and the *non-top spread* $\Delta(x) := t_{(2)}(x) - t_{(K)}(x) \geq 0$. For any $x$ with predicted index $M$ and confidence $p = q_M(x)$,

$$\mathbb{E}[\langle t \rangle_q - t_Y \mid X = x] \ \leq \ \left(p - r_X(x)\right) g(x) \ + \ \left(1 - r_X(x)\right) \Delta(x). \tag{28}$$

In particular, for binary classification ($K = 2$) one has $\Delta \equiv 0$ and equation 28 reduces to $\mathbb{E}[\langle t \rangle_q - t_Y \mid X] = (p - r_X)\, g$ (exact).

*Derivation of equation 28.* With $\eta_k(x) := \mathbb{P}(Y{=}k \mid X{=}x)$,

$$\mathbb{E}[\langle t \rangle_q - t_Y \mid X = x] = \sum_k (q_k - \eta_k)\, t_k = (p - r_X)\left(t_M - t_{(2)}\right) + \sum_{j \neq M} (q_j - \eta_j)\left(t_j - t_{(2)}\right).$$

Since $t_j - t_{(2)} \leq 0$ and $\sum_{j \neq M}(\eta_j - q_j)_+ \leq \sum_{j \neq M} \eta_j = 1 - r_X$, the last sum is $\leq (1 - r_X)\left(t_{(2)} - t_{(K)}\right) = (1 - r_X)\,\Delta$. $\square$

**Consequences and a mild spread control.** On a slice $A = \{m \in G\}$, assume the empirically checkable *spread control*

$$\Delta(x) \ \leq \ \Delta_G \ < \ \infty \qquad \text{for all } x \in A. \tag{29}$$

Then, combining equation 28 with Lemma 1,

$$\frac{d}{ds} L_{\mathrm{nll}}(h_s)\Big|_{s=1} = \mathbb{E}\big[\mathbf{1}_A(\langle t \rangle_q - t_Y)\big] \ \leq \ \mathbb{E}\big[\mathbf{1}_A\,(p - r_X)\, g\big] \ + \ \Delta_G\,\mu_A, \tag{30}$$

where $\mu_A := \mathbb{P}\{X \in A\}$. In the binary case $\Delta \equiv 0$ and equation 30 holds with equality.

**Two-slice *mismatch* under mild, empirically observed heterogeneity.** We next give conditions under which a single SMART-feasible local move reduces NLL yet *increases* smCE.

*Assumptions (empirically checkable).* Fix a *compact* margin slice $G \subset [m_{\min}, m_{\max}]$ and set $A := \{x : m(x) \in G\}$. Let $\gamma_{\min} := \inf_{x \in A} \frac{m(x)}{T(x)}$ and $\gamma_{\max} := \sup_{x \in A} \frac{m(x)}{T(x)}$ (finite and positive by $G$ compact and $T \in [T_{\min}, T_{\max}]$). Assume there exist disjoint compact intervals $J_{\mathrm{U}}, J_{\mathrm{O}} \subset (p_0, 1)$ with gap $\Delta > 0$ and constants $\rho_{\mathrm{U}}, \rho_{\mathrm{O}} > 0$ such that

$$r(p) - p \ \geq \ \rho_{\mathrm{U}} \quad \text{for } p \in J_{\mathrm{U}}, \qquad r(p) - p \ \leq \ -\rho_{\mathrm{O}} \quad \text{for } p \in J_{\mathrm{O}}.$$

Write $\mu_{\mathrm{U}} := \mathbb{P}\{p \in J_{\mathrm{U}},\, x \in A\}$, $\mu_{\mathrm{O}} := \mathbb{P}\{p \in J_{\mathrm{O}},\, x \in A\}$, $\mu_{\mathrm{gap}} := \mathbb{P}\{x \in A,\, p \notin J_{\mathrm{U}} \cup J_{\mathrm{O}}\}$, and assume additionally:

(a) (*bounded conditional density of $p$ on $A$*) the conditional distribution of $p$ given $X \in A$ has a density $f_{p|A}$ on $(0, 1)$ with $\|f_{p|A}\|_\infty \leq D_G < \infty$. In particular, for any interval $I \subset (0, 1)$, $\mathbb{P}\{p \in I,\, X \in A\} \leq D_G\,|I|$.

(b) (*slice-bounded advantage*) there exists $C_G < \infty$ with $p(x)\big(t_M(x) - \langle t(x) \rangle_{q(x)}\big) \leq C_G$ for $x \in A$ (e.g., it holds with $C_G := 2\,\mathrm{ess\,sup}_{x \in A} \|t(x)\|_\infty$ whenever $t$ is essentially bounded on $A$).

(c) the spread control equation 29 holds on $A$ with constant $\Delta_G$.

**Proposition A.2** (Two-slice mismatch: NLL $\downarrow$ but smCE $\uparrow$ (multi-class)). *Consider the* sharpening direction $s \uparrow 1$ *applied on the SMART-feasible set* $A = \{m \in G\}$. *If*

$$\rho_{\mathrm{U}}\,\gamma_{\min}\,\mu_{\mathrm{U}} \ > \ \gamma_{\max}\left(\rho_{\mathrm{O}}\,\mu_{\mathrm{O}} + \mu_{\mathrm{gap}}\right) \ + \ \Delta_G\,\mu_A, \tag{31}$$

*then $\frac{d}{ds}L_{\mathrm{nll}}(h_s)\big|_{s=1} < 0$ (NLL strictly decreases). Moreover, for any $c \in (0, \min\{1, \Delta\})$ there exists a 1-Lipschitz probe $\psi$ with $\psi \equiv 0$ on $J_{\mathrm{U}}$, $\psi \equiv -c$ on $J_{\mathrm{O}}$, and with transitions confined to a band whose $A$-mass is at most $\varepsilon > 0$, such that*

$$\frac{d}{ds}\mathbb{E}[(Y^\top - p_s)\psi(p_s)]\bigg|_{s=1} \geq c\,\underline{p}_{\mathrm{O}}\,(1-\overline{p}_{\mathrm{O}})\,\gamma_{\min}\,\mu_{\mathrm{O}} \,-\, (1+c)\,C_G\,\varepsilon, \tag{32}$$

*where $\underline{p}_{\mathrm{O}} := \inf J_{\mathrm{O}}$ and $\overline{p}_{\mathrm{O}} := \sup J_{\mathrm{O}}$. Choosing $\varepsilon < \frac{c\,\underline{p}_{\mathrm{O}}(1-\overline{p}_{\mathrm{O}})\gamma_{\min}}{(1+c)\,C_G}\mu_{\mathrm{O}}$ makes the right-hand side strictly positive. Because on $J_{\mathrm{O}}$ one has $(r(p) - p)\psi(p) \geq c\,\rho_{\mathrm{O}}$ while $\psi \equiv 0$ on $J_{\mathrm{U}}$, the signed functional at $s = 1$ obeys*

$$\mathbb{E}[(Y^\top - p)\psi(p)] \,=\, \mathbb{E}[(r(p) - p)\psi(p)] \,\geq\, c\,\rho_{\mathrm{O}}\,\mu_{\mathrm{O}} \,-\, c\,\varepsilon \,>\, 0, \tag{33}$$

*so a positive derivative implies a strict increase of its absolute value. Hence smCE strictly increases along $s \uparrow 1$.*

*Proof.* By equation 30 and splitting $A$ into the three regions,

$$\frac{d}{ds}L_{\mathrm{nll}}(h_s)\bigg|_{s=1} \leq \mathbb{E}\big[\mathbf{1}_{A \cap \{p \in J_{\mathrm{U}}\}}(p - r_X)g\big]$$
$$+ \mathbb{E}\big[\mathbf{1}_{A \cap \{p \in J_{\mathrm{O}}\}}(p - r_X)g\big]$$
$$+ \mathbb{E}\big[\mathbf{1}_{A \cap \{p \notin J_{\mathrm{U}} \cup J_{\mathrm{O}}\}}(p - r_X)g\big]$$
$$+ \Delta_G\,\mu_A.$$

On $A \cap \{p \in J_{\mathrm{U}}\}$, $g \geq \gamma_{\min}$ and $\mathbb{E}[p - r_X \mid p] = p - r(p) \leq -\rho_{\mathrm{U}}$, hence the contribution is $\leq -\rho_{\mathrm{U}}\gamma_{\min}\mu_{\mathrm{U}}$. On $A \cap \{p \in J_{\mathrm{O}}\}$, $g \leq \gamma_{\max}$ and $\mathbb{E}[p - r_X \mid p] \geq \rho_{\mathrm{O}}$, giving at most $\gamma_{\max}\rho_{\mathrm{O}}\mu_{\mathrm{O}}$. On the gap region, $|p - r_X| \leq 1$ and $g \leq \gamma_{\max}$, giving at most $\gamma_{\max}\mu_{\mathrm{gap}}$. This yields strict negativity under equation 31. For the probe, on $J_{\mathrm{O}}$ we have $\psi'(p) = 0$ and $-\psi(p) = c$, so by equation 26 and Lemma 2,

$$\frac{d}{ds}\mathbb{E}[(Y^\top - p_s)\psi(p_s)]\bigg|_{s=1,\,p \in J_{\mathrm{O}}} \geq c\,\underline{p}_{\mathrm{O}}\,(1-\overline{p}_{\mathrm{O}})\,\gamma_{\min}.$$

On $J_{\mathrm{U}}$ the contribution is 0 since $\psi \equiv 0$. On the transition band (of $A$-mass $\varepsilon$), $|\psi'| \leq 1$ and $|\psi| \leq c$, hence $|\psi'(p)(r_X - p) - \psi(p)| \leq (1+c)$ while $p(t_M - \langle t \rangle_q) \leq C_G$ on $A$ by (b). Thus the transition contribution is at most $(1+c)\,C_G\,\varepsilon$ in magnitude, giving equation 32. Finally, equation 33 holds since $\psi$ depends only on $p$ and $\mathbb{E}[Y^\top - p \mid p] = r(p) - p$. By (a), we can realize the 1-Lipschitz $\psi$ with linear ramps of total width at most $2c$, whence $\varepsilon \leq 2D_G c$; shrinking $c$ if needed makes the stated choice of $\varepsilon$ feasible. $\square$

**Lemma 3** (Small-$s$ realization for the mismatch). *Under Proposition A.2, there exists $s^\uparrow > 1$ arbitrarily close to 1 with*

$$L_{\mathrm{nll}}(h_{s^\uparrow}) \,<\, L_{\mathrm{nll}}(h) \qquad \text{and} \qquad \mathrm{smCE}(f_{h_{s^\uparrow}}) \,>\, \mathrm{smCE}(f_h).$$

*Proof.* $L_{\mathrm{nll}}(h_s)$ is $C^1$ at $s = 1$ by Lemma 1, with strictly negative derivative; hence $L_{\mathrm{nll}}(h_{s^\uparrow}) < L_{\mathrm{nll}}(h)$ for all $s^\uparrow > 1$ sufficiently close to 1. For smCE, fix the $\psi$ from Proposition A.2; then $F_\psi(s) := \mathbb{E}[(Y^\top - p_s)\psi(p_s)]$ is $C^1$ with $F_\psi(1) > 0$ and $F'_\psi(1) > 0$, so $|F_\psi(s^\uparrow)| > |F_\psi(1)|$ for all $s^\uparrow > 1$ close enough to 1. Since $\mathrm{smCE}(f_{h_s}) \geq |F_\psi(s)|$, it follows that $\mathrm{smCE}(f_{h_{s^\uparrow}}) > \mathrm{smCE}(f_h)$. $\square$

**Takeaway.** Along SMART-feasible local scalings of the temperature map $T(x) = h(m(x))$, Charbonnier–SoftECE continues to control smCE via equation 24, whereas NLL can be *locally improved* (decreased) while smCE *deteriorates* (increases) under mild, empirically checkable heterogeneity of confidence slices (Proposition A.2). The NLL directional formula is exact in binary classification; in multi-class settings the same conclusion holds under a weak spread control on non-top logits.

---

**Algorithm 1** SMART: Sample Margin-Aware Recalibration of Temperature

---

1: **Input:** Validation logits and labels $\{\mathbf{z}_i, y_i\}_{i=1}^{N_{\text{val}}}$, temperature network $h_\phi(\cdot)$
2: Compute margins: $m_i = z_{i,\max} - z_{i,\text{2nd}}$ for each $i \in \{1, \ldots, N_{\text{val}}\}$
3: Normalise: $\hat{m}_i = (m_i - \mu_m)/\sigma_m$ where $\mu_m = \frac{1}{N_{\text{val}}} \sum_i m_i$, $\sigma_m = \sqrt{\frac{1}{N_{\text{val}}} \sum_i (m_i - \mu_m)^2}$
4: **for** epoch $= 1, \ldots, N_{\text{epochs}}$ **do**
5:     Predict temperatures: $T_i = h_\phi(\hat{m}_i)$ for each $i$
6:     Scale logits: $\tilde{\mathbf{z}}_i = \mathbf{z}_i/T_i$ for each $i$
7:     Compute loss: $\mathcal{L}_i = \text{CharbonnierSoftECE}(\tilde{\mathbf{z}}_i, y_i)$ for each $i$ (Equation 6)
8:     Update: $\phi \leftarrow \phi - \eta \nabla_\phi \sum_{i=1}^{N_{\text{val}}} \mathcal{L}_i$ via SGD
9: **end for**
10: **Return:** Trained temperature network $h_\phi$

---

## B  THE USE OF LARGE LANGUAGE MODELS

During the preparation of this work, we utilized a Large Language Model (LLM) to assist with editorial refinement of the manuscript. The model's application was limited exclusively to improving textual quality and presentation, not for generating substantive research content. The LLM's contributions included:

- Enhancing sentence structure and paragraph organization to improve clarity, brevity, and scholarly tone.

- Identifying and correcting errors in grammar, spelling, and punctuation.

- Strengthening coherence and smoothing transitions throughout the text.

## C  RUNTIME EFFICIENCY

To verify the time efficiency of our method, we compare the inference time with baseline methods. The result is reported in Table 5. TS optimizes a single scalar temperature via a few gradient steps or closed-form updates, then applies this same factor to every logit, resulting in a negligible overhead (2.42 s). SMART yields a small per-sample inference cost and hence a modest total runtime (23.03 s). Logits are input into PTS's small neural network for each sample to predict a bespoke temperature, incurring a larger computational cost than SMART. CTS is the most expensive at more than 1 hour with the highest variance, as it conducts an exhaustive grid search for 5 epochs over a dense temperature grid (e.g. 0.1 -10) for each of the 1 000 classes, leading to $O(C \times G \times N)$ evaluations (classes $\times$ grid points $\times$ samples). The spline-based calibrator precomputes a monotonic mapping on the validation set and then applies a fast piecewise-linear transform at test time, yielding intermediate overhead. These differences illustrate the trade-off between expressive power and efficiency: TS is almost instantaneous, SMART adds only a small network-forward cost per sample, PTS trades per-sample flexibility for moderate cost, and CTS's brute-force search becomes prohibitive at scale.

Table 5: **Average Runtime (s) on ImageNet** over 10 runs on a ResNet-50 model.

| Method | TS | Spline | PTS | CTS | SMART |
|---|---|---|---|---|---|
| Runtime (s) | $2.42 \pm 0.1$ | $28.51 \pm 0.9$ | $1050.44 \pm 37.8$ | $5457.55 \pm 125.5$ | $23.03 \pm 0.41$ |

## D  THE PROPOSED SMART FRAMEWORK

This section presents the detailed algorithmic implementation of SMART, providing a step-by-step procedure for applying margin-based temperature scaling with soft-binned ECE optimization.

## E    FULL CALIBRATION PERFORMANCE

Full calibration performance for Table 1 is in Table 6.

Table 6: **Comparison of Post-Hoc Calibration Methods Using ECE (%, ↓, 15 bins) Across Various Datasets and Models** (mean ± std across 5 seeds). The best-performing method for each dataset-model combination is in bold, and our method is highlighted.

| Dataset | Model | Vanilla | TS | PTS | CTS | Spline | GC | ProCal | FC | SMART (ours) |
|---|---|---|---|---|---|---|---|---|---|---|
| CIFAR-10 | ResNet-50 | 4.34 | 1.38 ± 0.26 | 1.10 ± 0.21 | 0.83 ± 0.15 | 1.52 ± 0.03 | 1.37 ± 0.08 | 4.17 ± 0.12 | 1.66 ± 0.09 | **0.76 ± 0.02** |
| | Wide-ResNet | 3.24 | 0.93 ± 0.20 | 0.90 ± 0.19 | 0.81 ± 0.17 | 1.74 ± 0.01 | 0.89 ± 0.06 | 2.81 ± 0.11 | 1.12 ± 0.07 | **0.43 ± 0.05** |
| CIFAR-100 | ResNet-50 | 17.53 | 5.61 ± 1.39 | 1.96 ± 0.48 | 3.67 ± 0.88 | 3.48 ± 0.00 | 5.70 ± 0.15 | 9.71 ± 0.18 | 2.91 ± 0.12 | **1.37 ± 0.27** |
| | Wide-ResNet | 15.34 | 4.50 ± 0.62 | 1.96 ± 0.27 | 3.01 ± 0.42 | 3.76 ± 0.00 | 9.44 ± 0.16 | 9.44 ± 0.16 | 4.49 ± 0.14 | **1.80 ± 0.10** |
| ImageNet-1K | ResNet-50 | 3.65 | 2.17 ± 0.03 | 0.95 ± 0.36 | 2.17 ± 0.78 | 0.62 ± 0.18 | 2.44 ± 0.12 | 1.08 ± 0.14 | 1.71 ± 0.08 | **0.52 ± 0.12** |
| | DenseNet-121 | 2.53 | 1.85 ± 0.04 | 1.02 ± 0.46 | 1.86 ± 0.81 | 0.81 ± 0.35 | 2.20 ± 0.25 | 1.52 ± 0.21 | 1.35 ± 0.29 | **0.57 ± 0.03** |
| | Wide-ResNet | 5.43 | 2.89 ± 0.11 | 1.14 ± 0.24 | 3.27 ± 0.69 | 0.66 ± 0.10 | 3.66 ± 0.16 | 1.57 ± 0.10 | 1.62 ± 0.09 | **0.52 ± 0.07** |
| | Swin-B | 5.05 | 3.91 ± 0.07 | 1.05 ± 0.05 | 1.53 ± 0.08 | 0.88 ± 0.14 | 4.95 ± 0.17 | 1.00 ± 0.15 | 5.05 ± 0.06 | **0.46 ± 0.03** |
| | ViT-B-16 | 5.62 | 3.60 ± 0.19 | 1.23 ± 0.29 | 4.65 ± 1.02 | 0.91 ± 0.31 | 4.39 ± 0.25 | 0.97 ± 0.30 | 5.65 ± 0.06 | **0.48 ± 0.13** |
| | ViT-B-32 | 6.39 | 3.93 ± 0.02 | 1.27 ± 0.97 | 2.12 ± 1.59 | 0.81 ± 0.12 | 4.47 ± 0.13 | 0.88 ± 0.32 | 6.39 ± 0.06 | **0.71 ± 0.18** |
| ImageNet-C | ResNet-50 | 13.82 | 1.97 ± 0.02 | 1.12 ± 0.13 | 1.69 ± 0.20 | 5.61 ± 0.15 | 2.69 ± 0.11 | 5.79 ± 0.19 | 2.51 ± 0.13 | **0.62 ± 0.03** |
| | DenseNet-121 | 12.57 | 1.58 ± 0.00 | 1.19 ± 0.15 | 1.44 ± 0.19 | 5.18 ± 0.13 | 2.01 ± 0.09 | 9.88 ± 0.24 | 9.44 ± 0.31 | **0.63 ± 0.01** |
| | Swin-B | 12.03 | 5.82 ± 0.05 | 1.53 ± 0.00 | 3.05 ± 0.01 | 2.58 ± 0.21 | 6.92 ± 0.18 | 2.53 ± 0.12 | 5.18 ± 0.17 | **1.23 ± 0.04** |
| | ViT-B-16 | 8.28 | 5.24 ± 0.01 | 1.27 ± 0.05 | 2.76 ± 0.10 | 1.71 ± 0.22 | 5.95 ± 0.15 | 1.96 ± 0.14 | 5.37 ± 0.20 | **1.06 ± 0.02** |
| | ViT-B-32 | 7.69 | 5.10 ± 0.00 | 1.07 ± 0.08 | 2.97 ± 0.24 | 1.43 ± 0.24 | 6.40 ± 0.16 | 1.55 ± 0.11 | 5.50 ± 0.18 | **0.96 ± 0.01** |
| ImageNet-LT | ResNet-50 | 3.63 | 2.01 ± 0.02 | 0.99 ± 0.32 | 2.17 ± 0.68 | 0.56 ± 0.10 | 2.20 ± 0.17 | 1.12 ± 0.20 | 1.80 ± 0.23 | **0.56 ± 0.04** |
| | DenseNet-121 | 2.50 | 1.80 ± 0.06 | 1.20 ± 0.26 | 1.88 ± 0.41 | **0.79 ± 0.07** | 2.05 ± 0.11 | 1.79 ± 0.09 | 1.76 ± 0.50 | 0.81 ± 0.01 |
| | Wide-ResNet | 5.40 | 2.99 ± 0.05 | 1.21 ± 0.77 | 2.87 ± 1.79 | 0.81 ± 0.24 | 3.59 ± 0.18 | 1.28 ± 0.06 | 1.68 ± 0.10 | **0.53 ± 0.02** |
| | Swin-B | 4.69 | 3.98 ± 0.12 | 1.21 ± 0.45 | 1.50 ± 0.56 | 0.79 ± 0.17 | 4.79 ± 0.27 | 0.95 ± 0.16 | 4.82 ± 0.10 | **0.58 ± 0.01** |
| | ViT-B-16 | 5.58 | 3.73 ± 0.13 | 1.14 ± 0.47 | 1.43 ± 0.58 | 0.66 ± 0.05 | 4.34 ± 0.14 | 0.77 ± 0.14 | 5.72 ± 0.08 | **0.56 ± 0.14** |
| | ViT-B-32 | 6.28 | 3.98 ± 0.06 | 1.35 ± 0.41 | 2.12 ± 0.63 | 0.72 ± 0.23 | 4.76 ± 0.08 | 0.83 ± 0.12 | 6.26 ± 0.03 | **0.60 ± 0.11** |
| ImageNet-S | ResNet-50 | 22.32 | 2.06 ± 0.06 | 1.69 ± 0.27 | 1.48 ± 0.23 | 9.76 ± 0.22 | 1.99 ± 0.16 | 9.52 ± 0.31 | 12.58 ± 1.35 | **0.92 ± 0.09** |
| | DenseNet-121 | 20.13 | 1.67 ± 0.28 | 1.93 ± 0.19 | 1.16 ± 0.11 | 9.20 ± 0.32 | 1.77 ± 0.15 | 12.93 ± 0.23 | 22.67 ± 1.07 | **0.59 ± 0.25** |
| | Swin-B | 24.61 | 6.50 ± 0.05 | 1.53 ± 0.19 | 3.62 ± 0.45 | 8.66 ± 0.15 | 6.92 ± 0.35 | 8.05 ± 0.30 | 1.70 ± 0.06 | **1.26 ± 0.05** |
| | ViT-B-16 | 16.57 | 5.75 ± 0.08 | 1.33 ± 0.21 | 2.84 ± 0.43 | 5.70 ± 0.19 | 6.36 ± 0.29 | 5.67 ± 0.38 | 1.93 ± 0.18 | **0.98 ± 0.08** |
| | ViT-B-32 | 14.22 | 4.99 ± 0.15 | 1.67 ± 0.27 | 3.25 ± 0.50 | 4.07 ± 0.21 | 6.23 ± 0.16 | 4.44 ± 0.23 | 1.56 ± 0.09 | **0.87 ± 0.18** |

## F    CALIBRATION PERFORMANCE ON OTHER METRICS

### F.1    ACCURACY PERFORMANCE

**Accuracy Preservation Analysis**    Table 7 confirms that SMART achieves superior calibration while perfectly preserving classification accuracy—a fundamental advantage of post-hoc methods. Unlike CTS, which suffers accuracy drops up to 1.48 percentage points due to class-specific boundary alterations, or Spline's variable impacts on transformers, SMART's design ensures zero accuracy loss. By operating exclusively on the margin rather than full logit vectors, SMART focuses solely on confidence scaling without disturbing the relative ordering that determines predictions. This preservation holds even under severe distribution shifts like ImageNet-C and ImageNet-Sketch, where SMART simultaneously maintains base model accuracy while dramatically improving calibration. This dual guarantee makes SMART uniquely suitable for safety-critical applications requiring both correct predictions and reliable uncertainty estimates.

### F.2    AdaECE PERFORMANCE

This section provides an in-depth analysis of calibration performance using AdaECE across different datasets and model architectures, complementing the results presented in Section 4.2. Adaptive-ECE is a measure of calibration performance that addresses the bias of equal-width binning scheme of ECE. It adapts the bin-size to the number of samples and ensures that each bin is evenly distributed with samples. The formula for Adaptive-ECE is as follows:

$$\text{Adaptive-ECE} = \sum_{i=1}^{\mathbb{B}} \frac{|B_i|}{N} |I_i - C_i| \text{ s.t. } \forall i, j \cdot |B_i| = |B_j| \tag{34}$$

AdaECE offers a more rigorous assessment of calibration quality than standard ECE by adapting bin boundaries to ensure uniform sample distribution, preventing calibration errors from being masked in sparsely populated confidence regions. Table 8 presents comprehensive AdaECE results across all evaluated datasets and architectures. SMART consistently outperforms competing methods under this metric, achieving the lowest AdaECE on 24 of 26 dataset-architecture combinations.

Table 7: **Comparison of Classification Accuracy (%) Across Calibration Methods (Seed 1–5 Averaged).**

| Dataset | Model | Vanilla | TS | PTS | CTS | Spline | SMART |
|---|---|---|---|---|---|---|---|
| CIFAR-10 | ResNet-50 | 95.05% | 95.05% | 95.05% | 94.88% | 95.05% | 95.05% |
| | Wide-ResNet | 96.13% | 96.13% | 96.13% | 96.09% | 96.13% | 96.13% |
| CIFAR-100 | ResNet-50 | 76.69% | 76.69% | 76.69% | 76.38% | 76.69% | 76.69% |
| | Wide-ResNet | 79.29% | 79.29% | 79.29% | 79.28% | 79.29% | 79.29% |
| ImageNet-1K | ResNet-50 | 76.16% | 76.16% | 76.16% | 75.32% | 76.17% | 76.16% |
| | DenseNet-121 | 74.44% | 74.44% | 74.44% | 73.71% | 74.43% | 74.44% |
| | Wide-ResNet | 78.46% | 78.46% | 78.46% | 77.70% | 78.46% | 78.46% |
| | Swin-B | 83.17% | 83.17% | 83.17% | 82.80% | 83.17% | 83.17% |
| | ViT-B-16 | 81.12% | 81.12% | 81.12% | 79.64% | 80.86% | 81.12% |
| | ViT-B-32 | 75.95% | 75.95% | 75.95% | 75.14% | 75.94% | 75.95% |
| ImageNet-C | ResNet-50 | 19.16% | 19.16% | 19.16% | 19.34% | 19.16% | 19.16% |
| | DenseNet-121 | 21.25% | 21.25% | 21.25% | 21.36% | 40.83% | 21.25% |
| | Swin-B | 40.83% | 40.83% | 40.83% | 41.22% | 40.83% | 40.83% |
| | ViT-B-16 | 41.07% | 41.07% | 41.07% | 41.28% | 41.07% | 41.07% |
| | ViT-B-32 | 37.82% | 37.82% | 24.56% | 37.96% | 37.85% | 37.82% |
| ImageNet-LT | ResNet-50 | 76.04% | 76.04% | 76.04% | 75.43% | 76.04% | 76.04% |
| | DenseNet-121 | 74.34% | 74.34% | 74.34% | 73.88% | 74.40% | 74.34% |
| | Wide-ResNet | 78.39% | 78.39% | 78.39% | 77.67% | 78.40% | 78.39% |
| | Swin-B | 82.95% | 82.95% | 82.95% | 82.55% | 82.94% | 82.95% |
| | ViT-B-16 | 80.95% | 80.95% | 80.95% | 80.58% | 81.00% | 80.95% |
| | ViT-B-32 | 75.89% | 75.89% | 75.89% | 75.14% | 75.92% | 75.89% |
| ImageNet-S | ResNet-50 | 24.09% | 24.09% | 24.09% | 23.88% | 24.09% | 24.09% |
| | DenseNet-121 | 24.30% | 24.30% | 24.30% | 23.87% | 31.55% | 24.30% |
| | Swin-B | 31.54% | 31.54% | 31.54% | 31.65% | 31.55% | 31.54% |
| | ViT-B-16 | 29.37% | 29.37% | 29.37% | 29.51% | 29.39% | 29.37% |
| | ViT-B-32 | 27.77% | 27.77% | 27.77% | 27.76% | 27.75% | 27.77% |

Table 8: **Comparison of AdaECE Calibration Methods Using AdaECE($\downarrow$, %, 15bins) Across Various Datasets and Models (Seed 1–5 Averaged).**

| Dataset | Model | Vanilla | TS | PTS | CTS | Spline | SMART |
|---|---|---|---|---|---|---|---|
| CIFAR-10 | ResNet-50 | $4.33 \pm 0.0\%$ | $2.14 \pm 0.0\%$ | $0.83 \pm 28.6\%$ | $1.56 \pm 26.2\%$ | $2.14 \pm 1.1\%$ | $\mathbf{0.99 \pm 4.3}\%$ |
| | Wide-ResNet | $3.24 \pm 0.0\%$ | $1.71 \pm 0.0\%$ | $0.89 \pm 21.9\%$ | $1.47 \pm 19.7\%$ | $2.30 \pm 0.4\%$ | $\mathbf{0.50 \pm 12.2}\%$ |
| CIFAR-100 | ResNet-50 | $17.53 \pm 0.0\%$ | $5.66 \pm 0.0\%$ | $1.91 \pm 35.3\%$ | $3.43 \pm 32.0\%$ | $3.55 \pm 0.0\%$ | $\mathbf{2.27 \pm 25.2}\%$ |
| | Wide-ResNet | $15.34 \pm 0.0\%$ | $4.41 \pm 0.0\%$ | $1.69 \pm 13.0\%$ | $2.95 \pm 11.6\%$ | $3.95 \pm 0.1\%$ | $\mathbf{1.83 \pm 2.1}\%$ |
| ImageNet-1K | ResNet-50 | $3.68 \pm 1.3\%$ | $2.13 \pm 0.5\%$ | $0.92 \pm 44.1\%$ | $2.21 \pm 39.8\%$ | $0.81 \pm 28.7\%$ | $\mathbf{0.79 \pm 8.7}\%$ |
| | DenseNet-121 | $2.52 \pm 1.4\%$ | $1.74 \pm 1.8\%$ | $1.05 \pm 41.3\%$ | $1.78 \pm 38.0\%$ | $0.77 \pm 28.0\%$ | $\mathbf{0.65 \pm 10.2}\%$ |
| | Wide-ResNet | $5.31 \pm 0.3\%$ | $2.87 \pm 2.8\%$ | $1.04 \pm 20.6\%$ | $3.24 \pm 18.0\%$ | $\mathbf{0.83 \pm 36.3}\%$ | $0.87 \pm 14.3\%$ |
| | Swin-B | $4.86 \pm 0.6\%$ | $4.50 \pm 1.0\%$ | $1.05 \pm 4.6\%$ | $1.59 \pm 5.1\%$ | $1.04 \pm 5.3\%$ | $\mathbf{0.74 \pm 12.2}\%$ |
| | ViT-B-16 | $5.57 \pm 1.2\%$ | $4.10 \pm 2.3\%$ | $1.09 \pm 29.7\%$ | $4.85 \pm 27.4\%$ | $1.07 \pm 29.2\%$ | $\mathbf{0.79 \pm 15.4}\%$ |
| | ViT-B-32 | $6.41 \pm 0.4\%$ | $3.92 \pm 1.7\%$ | $1.27 \pm 71.9\%$ | $1.90 \pm 66.4\%$ | $0.96 \pm 15.3\%$ | $\mathbf{0.78 \pm 3.6}\%$ |
| ImageNet-C | ResNet-50 | $13.84 \pm 0.2\%$ | $2.02 \pm 1.7\%$ | $1.06 \pm 0.7\%$ | $1.76 \pm 0.6\%$ | $5.49 \pm 2.8\%$ | $\mathbf{0.74 \pm 8.0}\%$ |
| | DenseNet-121 | $12.57 \pm 0.1\%$ | $1.64 \pm 0.7\%$ | $1.17 \pm 9.9\%$ | $1.48 \pm 8.2\%$ | $2.57 \pm 7.9\%$ | $\mathbf{0.70 \pm 3.6}\%$ |
| | Swin-B | $11.98 \pm 0.1\%$ | $5.83 \pm 1.0\%$ | $1.58 \pm 0.0\%$ | $3.07 \pm 0.2\%$ | $5.13 \pm 2.3\%$ | $\mathbf{1.31 \pm 2.9}\%$ |
| | ViT-B-16 | $8.24 \pm 0.3\%$ | $5.25 \pm 0.9\%$ | $1.27 \pm 5.9\%$ | $2.77 \pm 5.3\%$ | $2.57 \pm 7.9\%$ | $\mathbf{1.09 \pm 4.0}\%$ |
| | ViT-B-32 | $7.66 \pm 0.2\%$ | $5.11 \pm 0.0\%$ | $1.07 \pm 4.3\%$ | $2.97 \pm 3.7\%$ | $1.45 \pm 16.8\%$ | $\mathbf{1.01 \pm 4.2}\%$ |
| ImageNet-LT | ResNet-50 | $3.54 \pm 0.9\%$ | $2.02 \pm 1.2\%$ | $0.92 \pm 35.5\%$ | $2.17 \pm 33.0\%$ | $0.71 \pm 20.7\%$ | $\mathbf{0.67 \pm 3.3}\%$ |
| | DenseNet-121 | $2.37 \pm 3.4\%$ | $1.74 \pm 2.1\%$ | $1.17 \pm 23.6\%$ | $1.86 \pm 21.3\%$ | $\mathbf{0.73 \pm 26.4}\%$ | $0.76 \pm 0.7\%$ |
| | Wide-ResNet | $5.22 \pm 0.4\%$ | $2.98 \pm 0.9\%$ | $1.22 \pm 62.4\%$ | $2.83 \pm 58.1\%$ | $0.79 \pm 18.1\%$ | $\mathbf{0.98 \pm 4.4}\%$ |
| | Swin-B | $4.69 \pm 0.6\%$ | $4.48 \pm 1.2\%$ | $1.43 \pm 19.1\%$ | $1.23 \pm 18.0\%$ | $0.95 \pm 6.7\%$ | $\mathbf{0.74 \pm 31.3}\%$ |
| | ViT-B-16 | $5.57 \pm 0.8\%$ | $4.18 \pm 2.9\%$ | $1.13 \pm 43.4\%$ | $1.06 \pm 40.1\%$ | $0.95 \pm 12.9\%$ | $\mathbf{0.85 \pm 15.1}\%$ |
| | ViT-B-32 | $6.26 \pm 0.6\%$ | $3.97 \pm 1.6\%$ | $1.30 \pm 31.1\%$ | $2.04 \pm 28.2\%$ | $0.86 \pm 26.5\%$ | $\mathbf{0.84 \pm 10.1}\%$ |
| ImageNet-S | ResNet-50 | $22.31 \pm 0.3\%$ | $2.01 \pm 2.9\%$ | $1.64 \pm 16.4\%$ | $1.51 \pm 14.7\%$ | $9.51 \pm 2.4\%$ | $\mathbf{0.90 \pm 15.8}\%$ |
| | DenseNet-121 | $20.15 \pm 0.5\%$ | $1.67 \pm 17.0\%$ | $1.93 \pm 9.6\%$ | $1.16 \pm 8.3\%$ | $8.7 \pm 1.92\%$ | $\mathbf{0.76 \pm 32.3}\%$ |
| | Swin-B | $24.62 \pm 0.0\%$ | $6.40 \pm 0.5\%$ | $1.53 \pm 12.2\%$ | $3.57 \pm 11.1\%$ | $9.06 \pm 4.2\%$ | $\mathbf{1.53 \pm 3.8}\%$ |
| | ViT-B-16 | $16.57 \pm 0.2\%$ | $5.62 \pm 0.7\%$ | $1.33 \pm 8.7\%$ | $2.98 \pm 7.3\%$ | $8.66 \pm 1.9\%$ | $\mathbf{1.08 \pm 4.3}\%$ |
| | ViT-B-32 | $14.19 \pm 0.3\%$ | $4.98 \pm 2.9\%$ | $1.66 \pm 16.0\%$ | $3.23 \pm 14.1\%$ | $5.64 \pm 3.3\%$ | $\mathbf{1.07 \pm 19.9}\%$ |

**CIFAR Performance Analysis.** SMART demonstrates exceptional calibration on CIFAR datasets in Figure 6, achieving the lowest AdaECE with notably stable variance compared to competitors. The key insight emerges when comparing CIFAR-10 to CIFAR-100: while global methods like TS suffer dramatic degradation as class count increases, SMART maintains robust performance. PTS shows

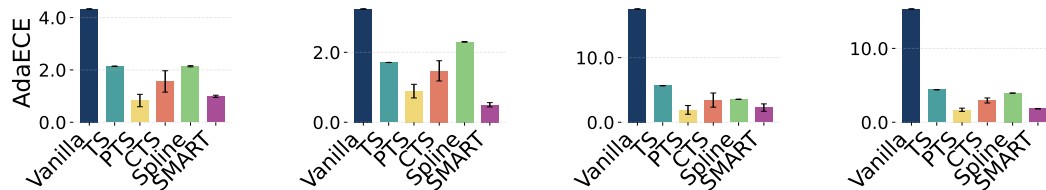

Figure 6: **AdaECE comparison on CIFAR datasets.** SMART consistently achieves superior calibration on both CIFAR-10 and CIFAR-100 across multiple architectures. From left to right are Cifar10 ResNet-50/Wide-ResNet, Cifar100 ResNet-50/Wide-ResNet.

competitive results but with substantially higher variance, indicating reliability issues. Spline struggles particularly with CIFAR-100's complex confidence landscape, revealing how non-parametric methods become less effective as classification complexity increases.

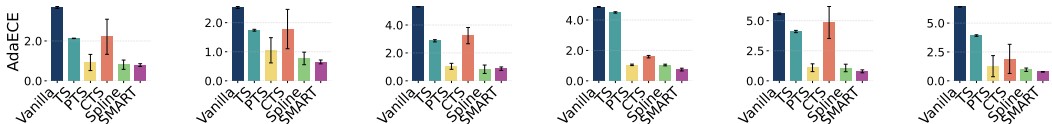

Figure 7: **AdaECE(↓, %, 15bins) comparison on ImageNet-1K.** SMART delivers consistent calibration across diverse architectures, from CNNs to vision transformers. From left to right are ResNet-50, DenseNet-121, Wide-ResNet, Swin-B, ViT-B-16, ViT-B-32.

**Large-Scale Classification on ImageNet.** In Figure 7, The ImageNet results reveal a crucial architectural insight: SMART maintains consistent performance across both CNN and transformer designs, while traditional methods like TS and CTS show pronounced degradation on transformers. This architectural robustness highlights SMART's ability to capture fundamental uncertainty signals through the margin regardless of model inductive biases. PTS exhibits extreme variance, confirming that high-dimensional parameterizations struggle with reliability when learning complex temperature mappings, particularly on large-scale datasets.



Figure 8: **AdaECE(↓, %, 15bins) comparison on ImageNet-C.** SMART maintains exceptional calibration under corruption, while Spline and TS-based methods demonstrate significant degradation. From left to right are ResNet-50, DenseNet-121, Swin-B, ViT-B-16, ViT-B-32.

**Robustness to Input Corruption.** As shown in Figure 8, SMART's resilience under corruption provides compelling evidence for the stability of decision boundary information. While Spline performs competitively on clean ImageNet, it deteriorates dramatically under corruption with values 5-7× higher than SMART. This collapse reveals a fundamental limitation: non-parametric methods overfit to validation distributions and fail when input characteristics change. SMART's focus on decision boundary uncertainty via the margin remains informative even when input distributions shift substantially.

**Long-Tailed Distribution Calibration.** As shown in Figure 9, The ImageNet-LT results reveal that class imbalance presents a fundamentally different calibration challenge than input corruption. Interestingly, Spline performs competitively here, suggesting non-parametric methods can handle statistical imbalances better than distributional shifts. However, CTS underperforms despite being

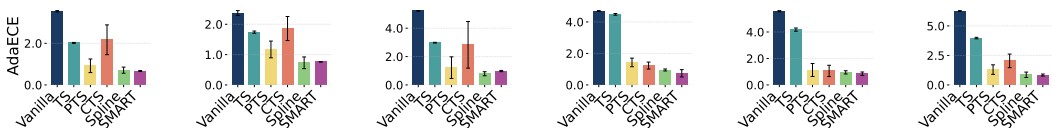

Figure 9: **AdaECE(↓, %, 15bins) comparison on ImageNet-LT.** SMART maintains strong calibration under long-tailed class distributions, particularly on CNN architectures. From left to right are ResNet-50, DenseNet-121, Wide-ResNet, Swin-B, ViT-B-16, ViT-B-32.

explicitly designed for per-class variations, demonstrating that simply applying different temperatures per class is insufficient for complex imbalanced scenarios.



Figure 10: **AdaECE(↓, %, 15bins) comparison on ImageNet-Sketch.** SMART maintains exceptional calibration under extreme domain shift, while Spline struggles significantly. From left to right are ResNet-50, DenseNet-121, Swin-B, ViT-B-16, ViT-B-32.

**Extreme Domain Shift Calibration.** The sketch-based domain shift represents the most challenging calibration scenario in Figure 10 , where SMART demonstrates its most dramatic advantage. Spline's collapse here reinforces the brittleness of non-parametric methods under distribution shifts, while SMART's consistent performance across all architectures provides strong evidence that margin information captures robust uncertainty signals that transcend specific input characteristics or domains.

# G COMPARISON OF VARIOUS TRAINING-TIME CALIBRATION METHODS ON OTHER METRICS

This section presents a comprehensive evaluation of SMART when combined with various training-time calibration methods across multiple metrics, extending the ECE analysis provided in Section 4.3. We examine SMART's performance using AdaECE, Classwise ECE (CECE), Negative Log-Likelihood (NLL), and classification accuracy.

## G.1 ACCURACY PRESERVATION

| Dataset | Model | NLL | | Brier Loss | | MMCE | | LS-0.05 | | FLSD-53 | | FL-3 | |
|---------|-------|------|------|------|------|------|------|------|------|------|------|------|------|
| | | base | ours | base | ours | base | ours | base | ours | base | ours | base | ours |
| CIFAR10 | ResNet-50 | 95.1 | 95.1 | 95.0 | 95.0 | 95.0 | 95.0 | 94.7 | 94.7 | 95.0 | 95.0 | 94.8 | 94.8 |
| | ResNet-110 | 95.1 | 95.1 | 94.5 | 94.5 | 94.6 | 94.6 | 94.5 | 94.5 | 94.6 | 94.6 | 94.9 | 94.9 |
| | DenseNet-121 | 95.0 | 95.0 | 94.9 | 94.9 | 94.6 | 94.6 | 94.9 | 94.9 | 94.6 | 94.6 | 94.7 | 94.7 |
| | Wide-ResNet | 96.1 | 96.1 | 95.9 | 95.9 | 96.1 | 96.1 | 95.8 | 95.8 | 96.0 | 96.0 | 95.9 | 95.9 |
| CIFAR100 | ResNet-50 | 76.7 | 76.7 | 76.6 | 76.6 | 76.8 | 76.8 | 76.6 | 76.6 | 76.8 | 76.8 | 77.3 | 77.3 |
| | ResNet-110 | 77.3 | 77.3 | 74.9 | 74.9 | 76.9 | 76.9 | 76.6 | 76.6 | 77.5 | 77.5 | 77.1 | 77.1 |
| | DenseNet-121 | 75.5 | 75.5 | 76.3 | 76.3 | 76.0 | 76.0 | 75.9 | 75.9 | 77.3 | 77.3 | 76.8 | 76.8 |
| | Wide-ResNet | 79.3 | 79.3 | 79.4 | 79.4 | 79.3 | 79.3 | 78.8 | 78.8 | 79.9 | 79.9 | 80.3 | 80.3 |

Table 9: **Comparison of Train-time Calibration Methods Using Accuracy(↑, %) Across Various Datasets and Models.** Results demonstrate that SMART preserves the original model accuracy across all training methods. Results are from the best run of 5 seeds.

**Accuracy Analysis** As shown in Table 9, SMART consistently preserves the classification accuracy of all base models across all training-time calibration methods. This is a critical property of post-hoc

calibration methods, as improving confidence estimates should not come at the cost of predictive performance. The perfect accuracy preservation is by design, as SMART's temperature scaling mechanism operates solely on the scaling of logits without altering their relative ordering, thus maintaining the same class predictions. This contrasts with some training-time methods that may involve trade-offs between accuracy and calibration quality during the model optimization process. The preservation of accuracy across diverse architectures and datasets further validates SMART's practical utility as a calibration method that can be safely applied in real-world scenarios where maintaining predictive performance is essential.

## G.2 ADAECE PERFORMANCE

| Dataset | Model | NLL | | Brier Loss | | MMCE | | LS-0.05 | | FLSD-53 | | FL-3 | |
|---------|-------|------|------|------|------|------|------|------|------|------|------|------|------|
| | | base | ours | base | ours | base | ours | base | ours | base | ours | base | ours |
| CIFAR10 | ResNet-50 | 4.33 | **0.80** | 1.74 | **1.01** | 4.55 | **0.67** | 3.88 | **2.18** | 1.56 | **0.45** | 1.95 | **0.48** |
| | ResNet-110 | 4.40 | **1.22** | 2.61 | **0.56** | 5.07 | **0.93** | 4.46 | **3.66** | 2.07 | **0.40** | 1.64 | **0.52** |
| | DenseNet-121 | 4.49 | **0.61** | 2.01 | **0.51** | 5.10 | **0.96** | 4.40 | **2.95** | 1.38 | **0.62** | 1.23 | **0.83** |
| | Wide-ResNet | 3.24 | **0.44** | 1.70 | **0.44** | 3.29 | **0.53** | 4.27 | **0.97** | 1.52 | **0.44** | 1.84 | **0.59** |
| CIFAR100 | ResNet-50 | 17.53 | **1.00** | 6.54 | **1.41** | 15.31 | **1.08** | 7.63 | **1.75** | 4.40 | **1.35** | 5.08 | **0.95** |
| | ResNet-110 | 19.06 | **1.67** | 7.73 | **0.93** | 19.13 | **1.98** | 11.07 | **2.72** | 8.54 | **0.93** | 8.65 | **1.22** |
| | DenseNet-121 | 20.99 | **2.23** | 5.04 | **1.02** | 19.10 | **1.73** | 12.83 | **1.96** | 3.54 | **0.93** | 4.14 | **0.97** |
| | Wide-ResNet | 15.34 | **1.55** | 4.28 | **0.97** | 13.16 | **1.12** | 5.13 | **2.11** | 2.77 | **0.75** | 2.07 | **1.15** |

Table 10: **Comparison of Train-time Calibration Methods Using AdaECE(↓, %, 15bins) Across Various Datasets and Models.** The best-performing method for each dataset-model combination is in bold, and our method (SMART) is highlighted. Results are from the best run of 5 seeds.

**AdaECE Analysis** The adaptive ECE results in Table 10 provide further validation of SMART's effectiveness when combined with various training-time calibration methods. AdaECE, which uses adaptive binning to ensure equal sample counts in each bin, offers a more robust calibration measure than standard ECE by eliminating potential biases from uneven confidence distributions. SMART consistently improves AdaECE across all training methods, with particularly dramatic improvements for models trained with NLL and MMCE, where we observe reductions of up to $18\times$ (17.53% $\rightarrow$ 1.00% for CIFAR-100 ResNet-50).

The most substantial AdaECE improvements occur on CIFAR-100, which has ten times more classes than CIFAR-10 and thus represents a more challenging calibration scenario. This suggests that SMART's effectiveness scales favorably with task complexity. Even for models already trained with calibration-oriented objectives like Focal Loss or FLSD, SMART provides further substantial improvements, indicating that its margin-based temperature adjustment captures complementary information to these training-time approaches. Notably, the combination of SMART with FLSD-53 achieves some of the lowest overall AdaECE values (e.g., 0.40% on CIFAR-10 ResNet-110), suggesting a particularly effective synergy between these methods.

## G.3 CLASSWISE ECE PERFORMANCE

| Dataset | Model | NLL | | Brier Loss | | MMCE | | LS-0.05 | | FLSD-53 | | FL-3 | |
|---------|-------|------|------|------|------|------|------|------|------|------|------|------|------|
| | | base | ours | base | ours | base | ours | base | ours | base | ours | base | ours |
| CIFAR10 | ResNet-50 | 0.91 | **0.43** | 0.46 | **0.40** | 0.94 | **0.51** | 0.71 | **0.51** | 0.42 | **0.37** | 0.43 | **0.38** |
| | ResNet-110 | 0.92 | **0.49** | 0.59 | **0.45** | 1.04 | **0.54** | 0.66 | **0.54** | 0.47 | **0.41** | 0.44 | **0.38** |
| | DenseNet-121 | 0.92 | **0.45** | 0.46 | **0.41** | 1.04 | **0.59** | 0.60 | **0.50** | 0.41 | **0.38** | 0.42 | **0.35** |
| | Wide-ResNet | 0.68 | **0.37** | 0.44 | **0.39** | 0.70 | **0.38** | 0.79 | **0.40** | 0.41 | **0.29** | 0.44 | **0.34** |
| CIFAR100 | ResNet-50 | 0.38 | **0.21** | 0.22 | **0.20** | 0.34 | **0.20** | 0.23 | **0.21** | 0.20 | **0.20** | 0.20 | **0.20** |
| | ResNet-110 | 0.41 | **0.20** | 0.24 | **0.21** | 0.42 | **0.21** | 0.26 | **0.20** | 0.24 | **0.20** | 0.24 | **0.21** |
| | DenseNet-121 | 0.45 | **0.23** | 0.20 | **0.20** | 0.42 | **0.23** | 0.29 | **0.21** | 0.19 | **0.20** | 0.20 | **0.20** |
| | Wide-ResNet | 0.34 | **0.19** | 0.19 | **0.19** | 0.30 | **0.19** | 0.21 | **0.20** | 0.18 | **0.18** | 0.18 | **0.18** |

Table 11: **Comparison of Train-time Calibration Methods Using Classwise ECE(↓, %, 15bins) Across Various Datasets and Models.** The best-performing method for each dataset-model combination is in bold, and our method (SMART) is highlighted. Results are from the best run of 5 seeds.

**CECE Analysis** Classwise ECE (CECE) provides insights into calibration performance at the individual class level rather than aggregated across all classes. The formula for classwise ECE is:

$$\text{Classwise-ECE} = \frac{1}{\mathcal{K}} \sum_{i=1}^{B} \sum_{j=1}^{\mathcal{K}} \frac{|B_{i,j}|}{N} |I_{i,j} - C_{i,j}| \tag{35}$$

where the calibration error is computed separately for each class $j$ across all bins $i$, then averaged across all $\mathcal{K}$ classes. This metric is particularly valuable for understanding whether calibration improvements are uniformly distributed across classes or concentrated in specific categories.

Table 11 demonstrates SMART's ability to improve per-class calibration across almost all training methods and architectures. The improvements are particularly prominent for models trained with NLL and MMCE, where CECE values are typically reduced by 50% or more after applying SMART (e.g., from 0.91% to 0.43% for CIFAR-10 ResNet-50). This substantial improvement suggests that SMART's margin-based temperature scaling effectively addresses class-specific miscalibration patterns that may arise during training with these standard objectives.

Interestingly, CECE values are consistently lower on CIFAR-100 compared to CIFAR-10 despite the higher class count, which contrasts with the pattern observed for ECE and AdaECE. This phenomenon occurs because CECE averages calibration errors across classes, and with 100 classes, individual class miscalibrations tend to average out more effectively than with only 10 classes. Additionally, the higher granularity of class divisions in CIFAR-100 may lead to more balanced per-class confidence distributions, making the averaging effect more pronounced.

For models already trained with calibration-oriented losses like FLSD-53 and FL-3, SMART provides more modest improvements in CECE, and in a few cases maintains the same level of performance. This suggests that these training-time methods are already effective at addressing per-class calibration issues through their specialized loss formulations that inherently consider class-wise balance. However, SMART can still provide complementary benefits in most scenarios, particularly for classes that may remain poorly calibrated even after specialized training procedures.

### G.4 NEGATIVE LOG-LIKELIHOOD PERFORMANCE

| Dataset | Model | NLL | | Brier Loss | | MMCE | | LS-0.05 | | FLSD-53 | | FL-3 | |
|---------|-------|-----|-----|------------|-----|------|-----|---------|-----|---------|-----|------|-----|
| | | Base | Ours | Base | Ours | Base | Ours | Base | Ours | Base | Ours | Base | Ours |
| CIFAR-10 | ResNet-50 | 41.2 | **19.7** | 18.7 | **18.4** | 44.8 | **21.0** | 27.7 | **27.7** | 17.6 | **17.1** | 18.4 | **17.9** |
| | ResNet-110 | 47.5 | **22.5** | 20.4 | **19.4** | 55.7 | **23.6** | 29.9 | **29.4** | 18.5 | **17.9** | 17.8 | **17.3** |
| | DenseNet-121 | 42.9 | **20.8** | 19.1 | **18.6** | 52.1 | **24.1** | 28.7 | **28.7** | 18.4 | **18.1** | 18.0 | **17.9** |
| | Wide-ResNet | 26.8 | **14.9** | 15.9 | **15.4** | 28.5 | **15.9** | 21.7 | **19.9** | 14.6 | **13.7** | 15.2 | **14.9** |
| CIFAR-100 | ResNet-50 | 153.7 | **105.3** | 99.6 | **99.5** | 125.3 | **100.7** | 121.0 | **120.1** | 88.0 | 88.4 | 87.5 | 88.1 |
| | ResNet-110 | 179.2 | **104.0** | 110.7 | **110.0** | 180.6 | **106.1** | 133.1 | **128.8** | 89.9 | **88.3** | 90.9 | **90.0** |
| | DenseNet-121 | 205.6 | **119.1** | 98.3 | 98.9 | 166.6 | **112.6** | 142.0 | **134.3** | 85.5 | 86.5 | 87.1 | 87.3 |
| | Wide-ResNet | 140.1 | **95.2** | 84.6 | 84.9 | 119.6 | **94.1** | 108.1 | **106.5** | 76.9 | 77.4 | 74.7 | 75.8 |

Table 12: **Comparison of Train-time Calibration Methods Using NLL(↓, %) Across Various Datasets and Models.** The best-performing method for each dataset-model combination is in bold, and our method (SMART) is highlighted. Results are from the best run of 5 seeds.

**NLL Analysis** NLL is a probabilistic metric that measures both calibration quality and discriminative power. Table 12 shows that SMART improves NLL for most models, with the most significant gains observed for NLL, MMCE, and LS-0.05 trained models. The improvements are particularly striking for CIFAR-10, where NLL is reduced by up to 60% after applying SMART (e.g., 41.22 → 19.70 for ResNet-50 with NLL).

However, a different pattern emerges for models trained with specialized losses like FLSD-53 and FL-3 on CIFAR-100, where SMART sometimes leads to slight increases in NLL despite improvements in calibration metrics like ECE and AdaECE. This suggests that these specialized training losses optimize directly for NLL-like objectives, creating a scenario where SMART's temperature scaling might slightly disturb the carefully optimized probability distributions. Nevertheless, the overall trend

across metrics indicates that SMART maintains or improves model performance in the vast majority of cases.

## H CALIBRATION PERFORMANCE UNDER SPECIFIC CORRUPTION TYPES

To provide deeper insights into SMART's robustness across different corruption scenarios, we examine the calibration error reduction achieved by various methods on individual corruption types in ImageNet-C. We analyze performance across two architectures (ResNet-50 and ViT-B/16) and two metrics (ECE and AdaECE), providing a comprehensive view of how different calibration approaches respond to specific distribution shifts. This granular analysis helps understand which corruption types pose the greatest calibration challenges and how architectural differences influence calibration robustness.

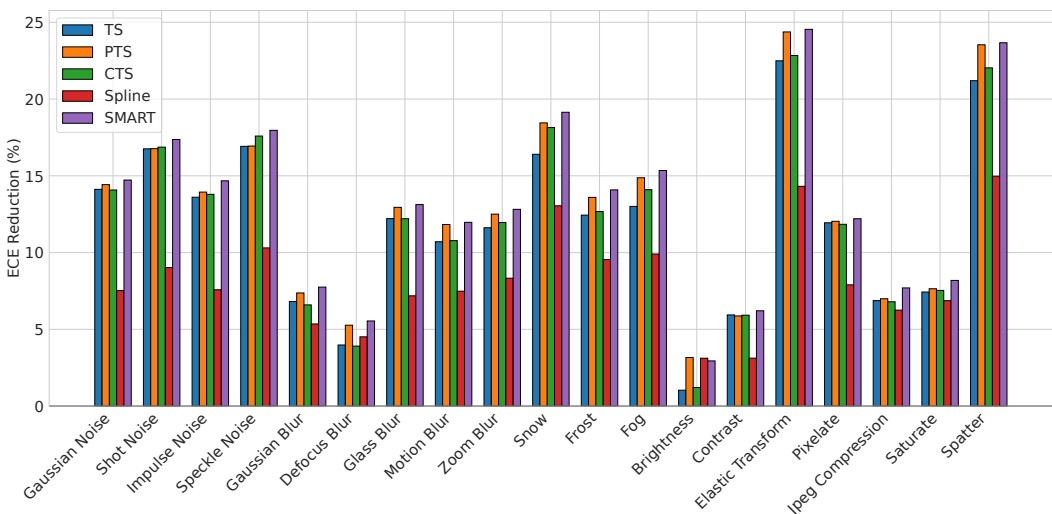

Figure 11: **ECE reduction(↑, %, 15bins) across corruption types for ResNet-50.** SMART consistently achieves superior calibration improvements across diverse corruption scenarios, demonstrating exceptional robustness to distribution shifts.

**ResNet-50 ECE Analysis** The corruption-specific analysis reveals that SMART demonstrates remarkable consistency, achieving the highest ECE reduction across most corruption categories with improvements often exceeding 20%. The inclusion of Spline calibration exposes a critical limitation of non-parametric methods: extreme brittleness under distribution shifts. While Spline achieves competitive results on certain corruptions like Snow, it completely fails on others such as Brightness and Contrast, highlighting how non-parametric approaches overfit to validation characteristics and break down when faced with novel corruptions.

This contrasts sharply with SMART's robust performance across all corruption types. The key insight is that SMART's margin indicator captures decision boundary information that remains meaningful regardless of input degradation type—whether geometric distortions, noise, or digital artifacts. Temperature Scaling and other global methods show predictable limitations on uniform corruptions, while parametric methods like PTS exhibit moderate consistency but still significant variability. SMART's sample-specific adaptation based on decision boundary information provides the most reliable calibration improvements, making it uniquely suitable for real-world scenarios where corruption characteristics are unpredictable.

**ResNet-50 AdaECE Analysis** The AdaECE results closely mirror the ECE patterns, confirming that SMART's calibration improvements are fundamental rather than evaluation artifacts. SMART achieves the highest reduction rates across most corruptions, with particularly strong performance on geometric distortions approaching 25% improvement. Spline's brittleness persists under adaptive binning—performing reasonably on weather corruptions but failing on uniform transforms, confirming

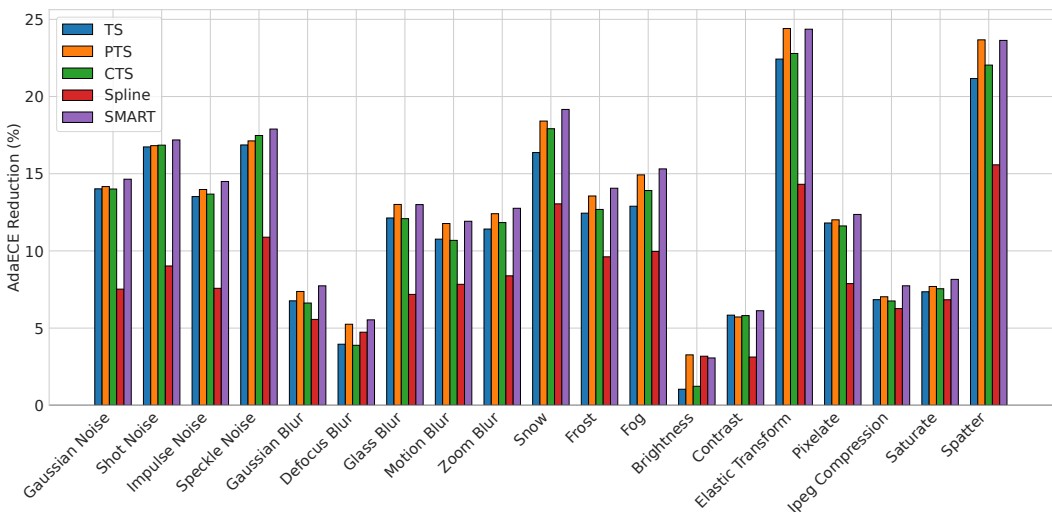

Figure 12: **AdaECE reduction(↑, %, 15bins) across corruption types for ResNet-50.** SMART maintains consistent superiority across corruption types under adaptive binning, confirming robust calibration improvements independent of evaluation methodology.

that its limitations stem from overfitting rather than evaluation methodology. The near-identical performance rankings across both metrics demonstrate that SMART's margin approach captures robust calibration signals regardless of how calibration quality is measured.

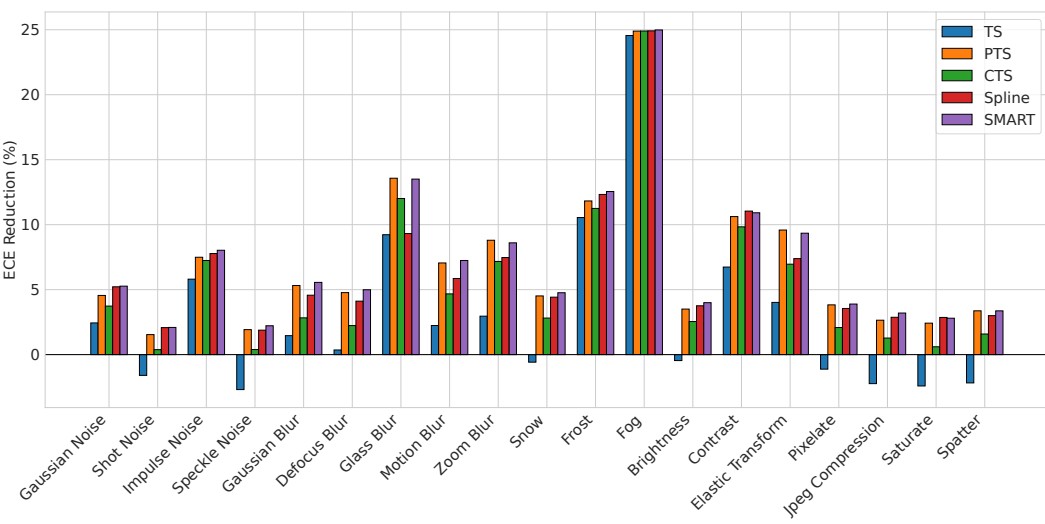

Figure 13: **ECE reduction(↑, %, 15bins) across corruption types for ViT-B/16.** Transformer architectures exhibit distinct calibration challenges under corruption, with global methods often failing while SMART maintains consistent improvements.

**ViT-B/16 ECE Analysis** The transformer results reveal striking architectural differences in calibration behavior under corruption. Most notably, Temperature Scaling frequently worsens calibration, showing negative improvements on multiple corruption types including Shot Noise, Speckle Noise, Snow, Brightness, Pixelate, Jpeg Compression, Saturate and Spatter. This demonstrates that transformers' attention mechanisms and different inductive biases make them fundamentally incompatible with global temperature adjustments under distribution shifts.

SMART maintains consistent positive improvements across all corruption types, though generally more modest than with ResNet-50. This architectural difference suggests that while transformers

are inherently better calibrated, they also present unique challenges that require more sophisticated approaches than global scaling. The convergence of all methods on Fog corruption (around 25% improvement) indicates that certain atmospheric corruptions create calibration conditions where architectural differences become less relevant.

A key insight emerges: the margin's decision boundary information remains meaningful across architectures, while global statistics become unreliable for transformers under corruption. PTS and CTS show more consistent improvements than TS, but SMART's sample-specific adaptation consistently outperforms all alternatives, confirming its architectural robustness.

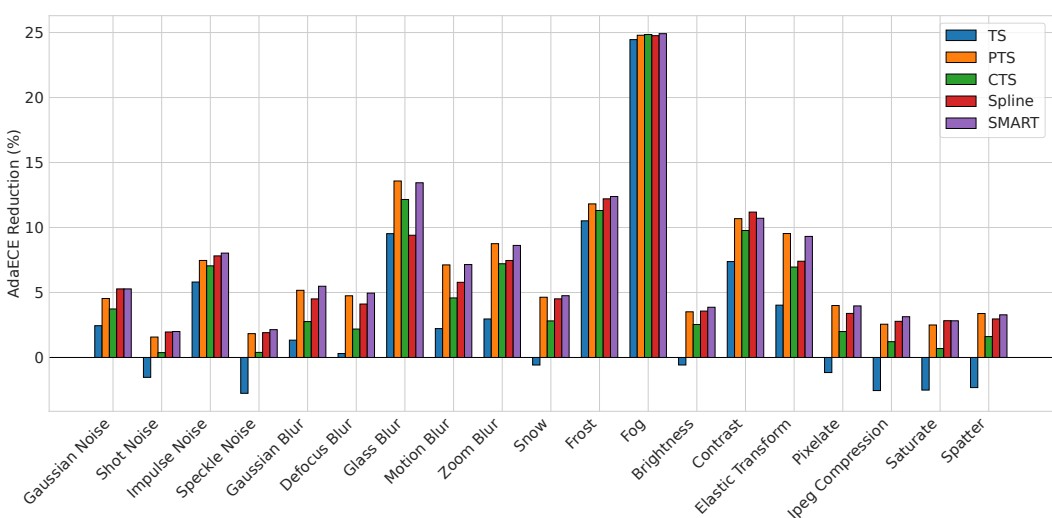

Figure 14: **AdaECE reduction(↑, %, 15bins) across corruption types for ViT-B/16.** Transformer calibration patterns remain consistent under adaptive binning, confirming architectural-specific calibration challenges and SMART's robustness.

**ViT-B/16 AdaECE Analysis**    The AdaECE results closely replicate the ECE patterns, confirming that transformer calibration behaviors are fundamental architectural characteristics rather than evaluation artifacts. Temperature Scaling's negative performance persists under adaptive binning, while SMART maintains consistent positive improvements across all corruption types. This metric independence demonstrates that SMART's margin approach captures robust decision boundary information that remains effective regardless of how calibration quality is measured.

## I   MARGIN PERSPECTIVE ON CALIBRATION

Traditional calibration analysis evaluates models from an overall perspective, potentially masking important sample-specific miscalibration patterns. By examining calibration behaviour across margin values, we uncover fundamental insights about how neural networks distribute confidence and validate our method visually.

Figure 15 demonstrates heterogeneity across margin groups. For ImageNet with ViT-B/16, whilst overall calibration appears near-perfect (Figure 15a), decomposing by margin reveals distinct patterns: low margin samples achieve good calibration (Figure 15c), whilst high margin samples show systematic under-confidence (Figure 15b). This pattern persists across different conditions, as shown in CIFAR-100 with ResNet-50 (Figures 15d and 15e), indicating that margin-based groupings reveal fundamental calibration characteristics transcending dataset-specific or architecture-specific behaviors.

**The Under-Confidence Paradox in High Margin Samples**    Perhaps the most counterintuitive finding emerges from examining high margin samples. Despite representing easy classifications with substantial separation between top predictions, these samples consistently exhibit under-confidence rather than expected over-confidence. High margin samples from ImageNet ViT-B/16 show systematic

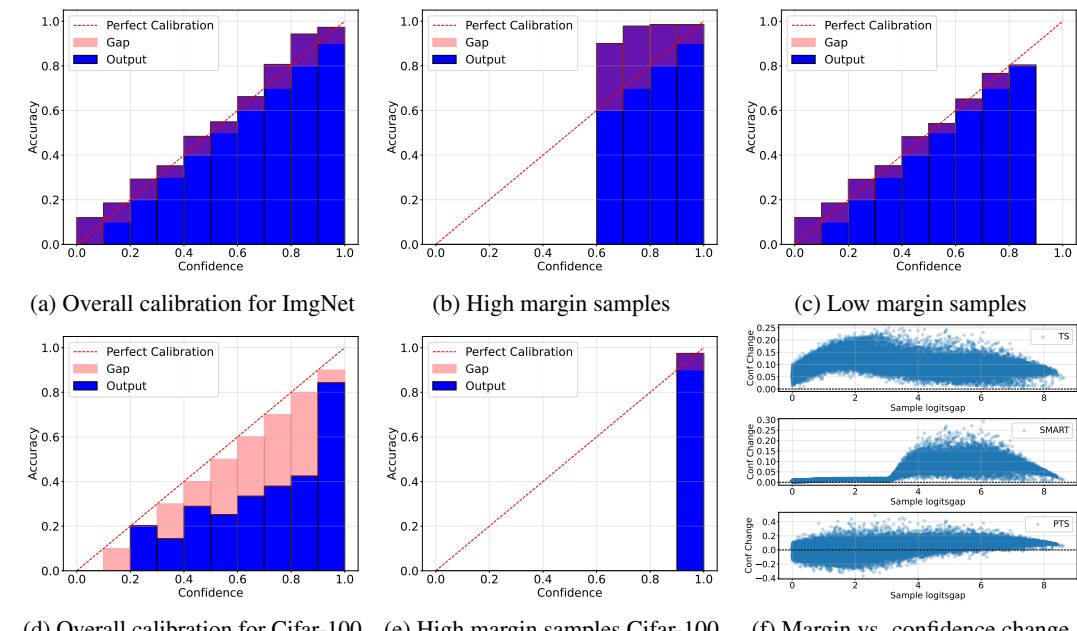

(a) Overall calibration for ImgNet    (b) High margin samples    (c) Low margin samples

(d) Overall calibration for Cifar-100   (e) High margin samples Cifar-100   (f) Margin vs. confidence change

Figure 15: **Margin reveals hidden calibration patterns across the confidence spectrum.** ImageNet ViT-B/16 shows near-perfect overall calibration (a) but reveals systematic under-confidence in high margin samples (b) and well-calibrated low margin samples (c). CIFAR-100 ResNet-50 demonstrates that even with overall over-confidence (d), high margin samples remain under-confident (e). Panel (f) shows SMART provides targeted adjustments whilst TS and PTS show suboptimal patterns.

under-confidence, with predicted confidence consistently lower than empirical accuracy (Figure 15b). This pattern persists even when overall model behaviour differs dramatically, as CIFAR-100 ResNet-50 maintains under-confidence in high margin samples despite overall over-confidence (Figure 15e).

**Method-Specific Failures from the Margin Perspective**    The confidence adjustment patterns in Figure 15f expose fundamental limitations in existing approaches. Temperature Scaling's uniform adjustment completely ignores heterogeneous calibration needs across margin groups, applying identical modifications regardless of sample characteristics. More critically, PTS makes substantial adjustments to low margin samples that already achieve good calibration and require minimal intervention. This unnecessary manipulation exemplifies how increased dimensionality introduces noise for precise temperature parameterisation. In contrast, SMART provides minimal adjustments to low margin samples that are already well-calibrated, whilst delivering targeted confidence increases to high margin samples suffering from under-confidence. This adaptive behavior emerges naturally from our lightweight margin-to-temperature mapping, demonstrating how principled architectural choices translate into appropriate calibration strategies.

## I.1 SENSITIVITY TO HYPERPARAMETERS $\lambda$ AND $\delta$

We examine the sensitivity of SMART's performance to the bandwidth parameter $\lambda$ and Charbonnier smoothing parameter $\delta$ in Equation equation 6. Tables 13 and 14 report ECE (15 bins) on ImageNet for ResNet-50 and ViT-B/16 across different $(\lambda, \delta)$ combinations.

The results demonstrate that performance remains stable within a reasonable range of values. For $\lambda \in \{0.01, 0.05, 0.10\}$, ECE varies by less than 0.2% across different $\delta$ choices, indicating robustness to the Charbonnier smoothing parameter. Larger values ($\lambda \geq 0.50$) lead to degraded performance due to over-localization of kernel weights, creating high variance in calibration estimates. Our choice of $\lambda = 0.05$ and $\delta = 0.001$ (highlighted rows) provides consistent performance across both CNN and transformer architectures, though the method is not particularly sensitive to $\delta$ within the range $[0.001, 0.100]$ when $\lambda$ is appropriately chosen.

Table 13: ECE (%, ↓, 15 bins) on ImageNet ResNet-50 for different $(\lambda, \delta)$ combinations.

| $\lambda\backslash\delta$ | 0.001 | 0.010 | 0.100 | 1.000 |
|---|---|---|---|---|
| 0.01 | 0.66 | 0.83 | 1.02 | 0.67 |
| 0.05 | 0.61 | 0.66 | 0.67 | 0.66 |
| 0.10 | 0.66 | 3.11 | 0.71 | 0.72 |
| 0.50 | 0.85 | 0.95 | 1.29 | 1.26 |
| 1.00 | 0.79 | 1.15 | 2.49 | 2.51 |

Table 14: ECE (%, ↓, 15 bins) on ImageNet ViT-B/16 for different $(\lambda, \delta)$ combinations.

| $\lambda\backslash\delta$ | 0.001 | 0.010 | 0.100 | 1.000 |
|---|---|---|---|---|
| 0.01 | 1.32 | 0.84 | 0.78 | 0.85 |
| 0.05 | 0.84 | 0.80 | 0.89 | 0.86 |
| 0.10 | 0.99 | 0.97 | 0.81 | 2.26 |
| 0.50 | 2.09 | 2.02 | 2.06 | 2.05 |
| 1.00 | 2.04 | 2.48 | 2.56 | 2.56 |

## J    ADDITIONAL ANALYSIS OF THE MARGIN–TEMPERATURE RELATIONSHIP

Figure 16 illustrates that the learned margin–temperature mapping is not constrained to be monotonic. For ImageNet ResNet-50, the mapping closely follows an increasing linear trend: samples with larger logit margins receive higher temperatures (softer probabilities), while low-margin samples are assigned temperatures closer to one. In contrast, on ImageNet with ViT-B/16 the mapping is clearly non-monotonic, with an approximately U-shaped dependence on the margin. This behavior indicates that the relationship between margin and miscalibration is architecture- and dataset-dependent; SMART adapts to these differences rather than enforcing a fixed monotone form, and understanding the underlying theoretical reasons is left for future work.

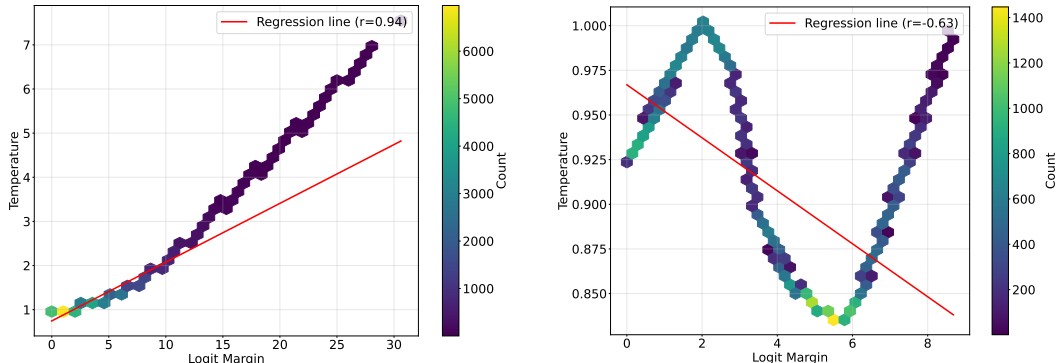

Figure 16: Empirical margin–temperature relationship learned by Left: ImageNet with a ResNet-50 backbone, where the mapping is approximately linear and monotone increasing (Pearson $r = 0.94$). Right: ImageNet with a ViT-B/16 backbone, where the mapping becomes non-monotonic with a pronounced U-shaped pattern (Pearson $r = -0.63$ for the best linear fit).

