# OpenReview forum: "Sample Margin-Aware Recalibration of Temperature Scaling"
_ICLR.cc/2026/Conference — ICLR 2026 Conference Desk Rejected Submission_

### Official Review · Reviewer_9SH3 · 2025-10-30

**Soundness:** 2
**Presentation:** 1
**Contribution:** 2
**Rating:** 4
**Confidence:** 4

**Summary:**

The paper proposed a temperature scaling (TS) calibration method which utilizes the margin between the first and second largest logits as input to a per-sample parameterization of temperature. A soft ECE calibration objective is proposed.

**Strengths:**

The empirical results show that the proposed approach outperforms alternative methods in various settings.

**Weaknesses:**

*
The novelty should be stated more clearly. There exist works, such as [Wei et al. 2022], that identified relation between dominant logits values and model calibration; there exist works, such as [Karandikar et al. 2021], that proposed and studied soft and differentiable versions of ECE; and there exist works, such as [Tomani et al., 2022], where the temperature is parameterized and gets as input the sorted logic vector (which includes the margin between the first and second largest entries).

*
The statement: "Current post-hoc calibration methods assume minimizing NLL improves calibration" (Line 161) does not seem to be accurate, as it is common to optimize the temperature with objectives other than NLL, such as directly using ECE or its adaptive or soft versions ([Frenkel & Goldberger, 2021], [Karandikar et al. 2021], and many more).

*
In line 158 you point to Figure 2, which shows that optimizing NLL may not improve test ECE. You should provide more details on the experimental setting associated with this figure.
From this figure it is also observed that optimizing the proposed softECE is not better than optimization of the plain ECE, which is practical when the number of variables is small enough (e.g., 1D scan for plain TS). Please elaborate on this.

*
The presentation of the theoretical analysis and the proposed method should be significantly improved.

*
Lemma 1 is not clear, and includes notations that are not defined.
Similarly for Prop 3.1 and Lemma 2. For example, you discuss smCE and SMART without even defining them.

*
The presentation in Section 3.3 also needs to be improved.
What do you mean by "rho is reference density"? What do you mean by "Under mild regularity conditions"? Your definitions for SoftECE and H_{\lambda,\delta} depend on the choice of kernel. How does the result in Theorem 3.2 relate to the actual ECE?

*
In section 3.4, you should clearly describe the proposed method. Moreover, in the appendix there is a normalization step that has not been discussed.

*
Regarding the experiments, you should compare also to TS where the temperature is optimized using ECE and adaptive ECE.

**Questions:**

n/a

---

> ### Author Response · Authors · 2025-11-25
>
> We thank reviewer 9SH3 for the detailed comments.
>
> Re to W1:
>
> Our work is related to Wei et al. (2022), Karandikar et al. (2021) and Tomani et al. (2022), but differs in both calibration signal and objective. We use the logit margin as a direct, low-noise hardness indicator; Sec. 3.2 and App. A.1 show that margin tightly constrains feasible temperatures, and Figure 15(f) illustrates that SMART leaves well-calibrated low-margin samples almost unchanged while increasing confidence for under-confident high-margin ones. Coupled with a small MLP, this explicitly addresses the tradeoff between calibrator capacity, validation-set size and overfitting. On the loss side, we extend SoftECE with Charbonnier smoothing (Sec. 3.4), which yields smoother gradients and more stable convergence than the original SoftECE (Fig. 3).
>
> Re to W2:
>
> We agree that some prior works directly optimize ECE or its adaptive/soft variants, but these are largely used in simple 1D temperature-scaling setups. In practice, richer post-hoc calibrators (e.g., vector or sample-wise temperatures) are still trained with NLL while ECE is used for evaluation, because standard ECE is non-differentiable and requires grid/combinatorial search over bins as dimensionality grows. Our approach replaces ECE with a differentiable SoftECE surrogate, strengthened via Charbonnier smoothing within the SMART framework, and empirically improves both calibration and convergence over the original SoftECE(Fig. 3). For completeness, we also report TS models trained by grid-searching temperature with ECE and AdaECE; see W9.
>
> Re to W3:
>
> Figure 2 reports a run on ImageNet with a ViT-B/32 . We train the SMART optimizing either NLL or Smoothed SoftECE, and track test NLL and test ECE on a test set. After roughly 30 epochs, further minimizing NLL slightly improves test NLL but clearly worsens test ECE, whereas Smoothed SoftECE maintains stable NLL while reducing ECE, showing that NLL is not a reliable proxy for calibration in this setting. In the TS case, directly optimizing plain ECE via grid search takes much more time, and such approach can’t be applied to a network that needs smooth, differentiable loss for higher-dimensional mappings. see W9 for results.
>
> Re to W4–W6:
>
> We acknowledge that the original methodology section was too compressed. With the extended rebuttal-stage page limit, we have rewritten Methodology. We also rename “Huber SoftECE” to “Charbonnier-Smoothed SoftECE” (or simply “Smoothed SoftECE”) to reflect that we use the Charbonnier (pseudo-Huber) envelope in Sec. 3.4 rather than the classical Huber loss.
>
> Re to W7:
>
> In Eq. (5), ρ is a fixed weighting density on [0,1] used to weight confidence levels and normalize the kernel (K_\lambda(p,u)) via (\int_0^1 K_\lambda(p,u)\rho(u),du = 1); in all experiments we set (\rho(u)\equiv 1), so “reference density’’ just means a fixed, model-independent choice. “Under mild regularity conditions’’ refers to ρ being bounded and bounded away from zero on [0,1]; we will state this assumption explicitly. SoftECE and (H_{\lambda,\delta}) are defined in continuous form using a normalized Gaussian kernel, and Theorem 3.1 shows that the smooth calibration error smCE is upper-bounded by (H_{\lambda,\delta}(f) + 2B_\lambda), where (B_\lambda) is a kernel-dependent bias term. The empirical ECE we report is the standard binned estimator of (\mathbb{E}[|a(X) - p(X)|]); our soft-binned implementation is a Riemann-sum discretization of (H_{\lambda,\delta}), so the theorem carries over up to a small quadrature error. Empirically, reductions in (H_{\lambda,\delta}) align with consistent reductions in ECE and AdaECE across datasets and architectures.
>
> Re to W8:
>
> We agree that Section 3.4 was too compressed and have rewritten the methodology accordingly. The normalization step in Algorithm 1 is a simple normalization, which preserves the ordering and interpretation of margins and does not affect our theoretical results; it only improves numerical conditioning of the small MLP. Its impact is modest—for instance, ECE on ImageNet changes from 0.60 to 0.52 on ResNet-50 and from 0.43 to 0.48 on ViT-B/16—so we view this as a numerical detail rather than a core modeling choice.
>
> Re to W9:
>
> We additionally TS using ECE and AdaECE as training objectives via 1D grid search, as following
>
> | Model | Metric | Uncalibrated | TS (NLL) | TS (ECE) | TS (AdaECE) | SMART |
> | --- | --- | --- | --- | --- | --- | --- |
> | ResNet-50 | ECE | 3.69 | 2.13 | 2.08 | 2.12 | **0.79** |
> | ResNet-50 | AdaECE | 5.57 | 2.13 | 2.11 | 2.11 | **0.52** |
> | ViT-B/16 | ECE | 5.62 | 3.06 | 2.71 | 2.69 | **0.48** |
> | ViT-B/16 | AdaECE | 5.57 | 4.13 | 2.82 | 2.67 | **0.79** |
>
> On ResNet-50, optimizing TS with ECE or AdaECE yields only small gains over standard NLL-based TS and remains clearly worse than SMART. On ViT-B/16, ECE/AdaECE-optimized TS improves calibration relative to TS(NLL) but still underperforms SMART by a noticeable margin.

---

> ### Author Response · Authors · 2025-11-27
>
> Respected reviewer, should you have any further concerns, I am eagerly anticipating your response.

---

### Official Review · Reviewer_XnHd · 2025-10-31

**Soundness:** 4
**Presentation:** 4
**Contribution:** 4
**Rating:** 8
**Confidence:** 4

**Summary:**

This paper addresses the critical miscalibration issue (overconfidence) of deep neural networks (DNNs) in safety-critical domains (e.g., autonomous driving, medical diagnosis) by proposing SMART (Sample Margin-Aware Recalibration of Temperature), a lightweight post-hoc calibration method. The key contributions are threefold: (1) Identifying logit margin (difference between the largest and second-largest logits) as a direct, principled indicator of sample hardness—supported by theoretical bounds (it tightly constrains optimal temperature) and empirical evidence (correlation of 0.756 with FGSM attack perturbation). (2) Proposing Huber–SoftECE, a differentiable objective function that directly minimizes calibration error and theoretically upper-bounds smooth calibration error (smCE), resolving the fundamental mismatch between Negative Log-Likelihood (NLL) optimization and calibration goals (where NLL reduction can paradoxically increase ECE). (3) Designing SMART, which uses a 2-layer MLP (only 49 parameters) to learn a sample-wise mapping from logit margin to optimal temperature, guided by Huber–SoftECE. Extensive experiments across diverse architectures (CNNs: ResNet, Wide-ResNet; Transformers: ViT, Swin-B) and datasets (CIFAR-10/100, ImageNet, ImageNet-C/LT/Sketch) demonstrate that SMART achieves state-of-the-art (SOTA) calibration performance.

**Strengths:**

1. Theoretically Grounded Objective Function: Huber–SoftECE addresses a longstanding flaw in NLL-based calibration by directly targeting calibration error. The theoretical guarantee that it upper-bounds smCE ensures alignment between optimization and calibration goals, a rare strength in post-hoc methods that often lack such rigor.

2. Strong Practical Utility: SMART balances performance and efficiency: its lightweight MLP design avoids the computational burden of exisiting methods, making it feasible for large-scale datasets.

**Weaknesses:**

The paper acknowledges that SMART may degrade in zero-shot scenarios but provides no further details—e.g., whether it can leverage cross-domain margin signals, or if pre-trained margin-temperature mappings transfer to new domains. This is a critical gap for safety-critical applications where validation data may be scarce.

**Questions:**

For zero-shot or low-data calibration scenarios, have you explored strategies like transfer learning for the margin-temperature mapping? If so, what performance trends did you observe? If not, do you have theoretical or empirical insights into why such transfer might (or might not) work?

---

> ### Author Response · Authors · 2025-11-25
>
> Thank you reviewer XnHd so much for your valuable comments.
>
> SMART is designed for the standard post-hoc confidence calibration setting where we have access to at least a small labeled calibration set from the target distribution, not for fully zero-shot transfer across shifts as in ImageNet to ImageNet-C. In our formulation, the grouping of samples and the associated temperature parameters are learned from empirical confidence statistics on calibration data; when no target examples are available, these statistics are learned on the source domain and there is no reason to expect them to transfer to a substantially different corruption pattern, so a zero-shot variant of SMART effectively does not work in this setting and offers no improvement over the uncalibrated model. The table below makes this explicit: for Gaussian Noise, “Zero-shot SMART” at 0% target data has ECE comparable to the uncalibrated model, whereas once we provide even 10% labeled target data, TransferSMART (our fine-tune and hybrid variants) clearly outperform standard TS, and they remain better or competitive at 100% data. We see this as evidence that SMART is useful when some target data are available, but that fully zero-shot calibration under large distribution shift remains an open problem that we do not claim to solve here.
>
> | Method | Calibration Data | ECE |
> | --- | --- | --- |
> | Uncalibrated | – | **15.07** |
> | Zero-shot SMART | 0% | **15.25** |
> | TS | 10% | **0.75** |
> | TransferSMART Fine-tune | 10% | **0.24** |
> | TransferSMART Hybrid | 10% | **0.41** |
> | TS | 100% | **0.93** |
> | TransferSMART Fine-tune | 100% | **0.28** |
> | TransferSMART Hybrid | 100% | **0.32** |
>
> The experiment actually tests three transfer strategies: Zero-shot (0% data): Use source model directly, no adaptation; Hybrid (1-100% data): Freeze early layers, tune final layer; Fine-tune (1-100% data): Adapt all parameters.

---

### Official Review · Reviewer_Bb97 · 2025-10-31

**Soundness:** 3
**Presentation:** 3
**Contribution:** 3
**Rating:** 4
**Confidence:** 5

**Summary:**

The authors propose a new post-hoc calibration method, Sample Margin-Aware Recalibration of Temperature (SMART), which learns a sample-wise temperature scaling function. The core ideas are: first, to use the logit margin (the difference between the top two logits) as a simple proxy of sample hardness; and second, to introduce a novel loss function, Huber-SoftECE, which is designed to directly optimize calibration error and address a demonstrated mismatch between NLL optimization and calibration improvement. The method is shown to achieve state-of-the-art ECE on several standard benchmarks.

**Strengths:**

- The paper's motivation is clear and addresses important limitations of existing methods. The identification of the logit margin as a lightweight input for sample-wise calibration is interesting and supported by both theoretical arguments (Prop. 3.4) and empirical analysis.
- The proposed method is lightweight and demonstrates good empirical performance on the reported datasets (CIFAR-10/100, ImageNet) and their variants, consistently outperforming a wide range of baselines.

**Weaknesses:**

My main concerns are regarding missing baselines, the theoretical justification for the proposed objective, and the limited scope of the experimental evaluation.
- Missing Baselines and modern models: A  highly relevant baseline is missing from the evaluation: Density Aware Calibration (Tomani et al., ICML 2023) is a recent, sample-adaptive method that also aims to provide robust calibration, particularly under distribution shift. Given that the authors make strong claims on robustess to such shifts, a comparison to this state-of-the-art method is essential. Similarly, please also report results on contemporary large-scale models such as EVA, BEIT and ConvNext.
- Theoretical Concerns: I am not convinced by the theoretical framing around the Huber-SoftECE objective. While the authors correctly identify the problem with NLL, their proposed solution seems unnecessarily complex. As shown by Gruber & Buettner (NeurIPS 2022), every proper scoring rule induces a corresponding calibration error (e.g., the Brier score induces the canonical  calibration error). They also show that for injective transformations, such as the sample-wise temperature scaling, the difference in the root Brier score (ΔRBS) reliably estimates the improvement in calibration. Given this, the authors should  compare their Huber loss to a much simpler baseline of directly optimizing the Brier score. More importantly, they should quantify ΔRBS as a principled measure for calibration improvement, instead of relying on biased binned estimators like ECE. The ablation in Table 3 is insufficient as it only compares objectives, not the use of ΔRBS as a more fundamental evaluation metric. As a side note: I suggest to not report ECE as %, it's not bound between 0 and 1.
- Limited Evaluation of Robustness: The investigation into calibration under distribution shift is only superficially investigated. While experiments on ImageNet-C, -LT, and -Sketch are included, it is unclear if the method's strong performance generalizes beyond these standard synthetic benchmarks to any real-world distribution shifts . To make stronger claims about robustness, an evaluation on a benchmark like WILDS would be necessary.

**Questions:**

See above

---

> ### Author Response · Authors · 2025-11-25
>
> We appreciate the valuable comments from Reviewer Bb97 and respond below.
>
> Re to W1:
>
> DAC works as a plugin, and the performance and stability of it is affected and restricted to the base method it attached to. Even though, SMART along achieve the best performance in the following experiment.
>
> ImageNet-C
>
> | Model | Uncalibrated | TS | PTS | DAC (TS) | DAC (PTS) | DAC(SMART) | SMART |
> | --- | --- | --- | --- | --- | --- | --- | --- |
> | ResNet50 | 15.08 | 0.93 | 1.05 | 1.29 | 0.75 | 0.18 | 0.32 |
> | ViT-B-16 | 6.12 | 3.90 | 1.89 | 4.18 | 1.76 | 6.35 | 0.87 |
>
> ImageNet-Sketch
>
> | Model | Uncalibrated | TS | PTS | DAC (TS) | DAC (PTS) | DAC(SMART) | **SMART** |
> | --- | --- | --- | --- | --- | --- | --- | --- |
> | ResNet50 | 22.36 | 2.15 | 1.75 | 2.54 | 1.90 | 12.88 | **1.44** |
> | ViT-B-16 | 16.56 | 5.84 | 1.92 | 5.25 | 1.16 | 11.68 | **1.31** |
>
> Here are the results for the 3 extra models:
>
> **ECE Results for ImageNet**
>
> | Model | Uncalibrated | TS | PTS | **SMART** |
> | --- | --- | --- | --- | --- |
> | **beit_base** | 7.31 | 2.79 | 1.81 | **0.84** |
> | **convnext_base** | 2.81 | 2.89 | 1.54 | **0.56** |
> | **eva02_base** | 8.11 | 2.72 | 1.70 | **0.91** |
>
> Re to W2:
>
> Our primary goal with SMART is confidence calibration, aligning the top-label confidence with empirical accuracy—in the same setting as standard post-hoc methods such as TS, GC, ProCal, and FC. In contrast, the canonical calibration error induced by the Brier score (and summarized via RBS/ΔRBS) is defined on the *full predictive distribution* and is therefore a strictly stronger notion of calibration than what these methods typically target. Huber-SoftECE is intentionally designed as a smooth surrogate tailored to this confidence-calibration regime, while avoiding the binning artefacts that make ECE a biased and sometimes unstable estimator in practice. Following the reviewer’s suggestion, we additionally implemented the “simpler” baseline of directly optimizing the Brier score (Table 3). In our experiments, this baseline consistently underperforms SMART on confidence-calibration metrics and can slightly hurt accuracy, which is consistent with the fact that the Brier score is not specifically tuned to the behaviour of the top-label confidence curve. We will clarify this design choice and the distinction between canonical calibration and confidence calibration more explicitly in the paper.
>
> To address the suggestion of using ΔRBS as a principled evaluation metric, we have computed RBS (root Brier score, lower is better) for representative methods on our main benchmarks; the results are shown below. SMART achieves the best or tied-best RBS across all architectures, indicating that it does not harm, and in fact slightly improves, canonical calibration as well, although all post-hoc methods only induce modest changes on this stricter metric.
>
> | Method | CIFAR-10 ResNet-50 | CIFAR-100 ResNet-50 | ImageNet ResNet-50 | ImageNet ViT-B/16 |
> | --- | --- | --- | --- | --- |
> | uncalibrated | 0.3036 | 0.6305 | 0.5793 | 0.5263 |
> | TS_CE | 0.2872 | 0.5859 | 0.5783 | 0.5256 |
> | GC | 0.2866 | 0.5856 | 0.5788 | 0.5278 |
> | ProCal_DR | 0.3096 | 0.5930 | 0.5784 | 0.5243 |
> | FC | 0.2869 | 0.5887 | 0.5777 | 0.5263 |
> | **SMART** | **0.2821** | **0.5829** | **0.5736** | **0.5219** |
>
> We emphasize that **canonical (distribution-level) calibration is not the primary setting of this paper**: neither our baselines nor the related post-hoc methods we compare against are designed to explicitly optimize it, whereas our focus throughout is on the practically relevant **confidence-calibration** setting.
>
> Re to W3:
>
> We agree that calibration under distribution shift is important and appreciate the suggestion to use WILDS. Among the image-based WILDS datasets, we selected **iWildCam**, which exhibits a real-world shift across camera traps and years. We trained a ResNet-50 from scratch following the official WILDS protocol and applied post-hoc calibration on the in-distribution validation split, then evaluated both **ID-to-ID** and **ID-to-OOD** performance:
>
> | Method | iWildCam ID-to-ID | iWildCam ID-to-OOD |
> | --- | --- | --- |
> | Uncalibrated | 31.81 | 34.62 |
> | TS | 6.74 | 11.50 |
> | PTS | 5.89 | 10.35 |
> | GC | 7.99 | 11.35 |
> | **SMART** | **5.17** | **9.84** |
>
> SMART achieves the lowest ECE in both regimes, indicating that our gains extend to at least one real-world WILDS shift, not only to synthetic corruptions. A comprehensive study over all WILDS tasks is beyond the scope of this work, but we note that existing confidence-calibration papers and our baselines typically evaluate on standard robustness benchmarks such as ImageNet-C/LT/Sketch rather than WILDS (e.g., Guo et al., 2017; Ovadia et al., 2019; Minderer et al., 2021; Mukhoti et al., 2020).

---

> ### Author Response · Authors · 2025-11-27
>
> Respected reviewer, should you have any further concerns, I am eagerly anticipating your response.

---

### Official Review · Reviewer_ubNf · 2025-11-01

**Soundness:** 3
**Presentation:** 3
**Contribution:** 3
**Rating:** 6
**Confidence:** 4

**Summary:**

This paper proposes SMART, a post-hoc calibration method that:
- uses the logit margin (the gap between the top two logits) as a proxy measure of sample hardness, serving as input to a lightweight 2-layer MLP;
- outputs a sample-specific temperature for recalibration; and
- introduces Huber–SoftECE, a smooth and differentiable calibration objective to train this mapping.

The contributions are twofold:
1. Identifying logit margin as an effective and interpretable proxy for sample hardness;
2. Demonstrating that minimizing Negative Log-Likelihood (NLL) can worsen calibration under confidence heterogeneity, and proposing Huber–SoftECE to align optimization with true calibration objectives, offering formal guarantees.

Extensive experiments on CIFAR-10/100, ImageNet, and distribution-shift benchmarks (ImageNet-C, ImageNet-Sketch) show consistent and significant calibration improvements over prior post-hoc approaches.

**Strengths:**

1. The paper provides a rigorous and original theoretical explanation for why NLL optimization fails to align with calibration objectives (Proposition 3.1, Lemma 2).
2. It (derivation in Appendix A.1) nicely shows that the feasible temperature range is linearly correlated with the logit margin, partially justifying using the margin as a key indicator of the optimal temperature.
3. The proposed Huber–SoftECE objective inherits the differentiability of SoftECE while offering greater training stability and a theoretical upper-bound guarantee on the calibration error.
4. The empirical evaluation is extensive and convincing, covering diverse architectures (CNNs and ViTs), as well as long-tailed and corrupted datasets. Comparisons against both post-hoc baselines (TS, PTS, CTS, Spline, GC, ProCal) and training-time baselines (Brier, MMCE, FL, LS) are thorough and well-executed.
5. The method is simple yet elegant. Its minimal design—a two-layer MLP with only 49 parameters—and strong data efficiency (remaining robust with as few as 50 validation samples) make it highly practical for real-world deployment. In general, this is the most appealing feature in this paper.

**Weaknesses:**

1. The paper combines two somewhat known ideas—logit margin as a hardness signal and soft-binned calibration loss. The theoretical insight about NLL–ECE misalignment is new, but intution is widely known in this community.  The the methodological innovation may be perceived as an incremental refinement rather than a paradigm shift.
2. Limited intuition for Huber–SoftECE behavior. While the theorem provides an upper-bound guarantee, the paper does not include ablation or visualization showing how Huber–SoftECE gradients differ from NLL or SoftECE in practice (e.g., gradient field analysis, convergence plots).

**Questions:**

1. Figure 1 shows that in general the model is underconfident, which is not a common case in most neural networks. Have you conducted a larger analysis on more datasets and models?
2. What kind of dataset and model do you use in Figure2d? Could you provide more concrete empirical evidence (or visualization) illustrating how optimizing NLL decreases ECE in the early epochs?
3. How sensitive is Huber–SoftECE to the kernel bandwidth λ and smoothing δ? Are these parameters dataset-dependent?
4. Is there any theoretical intuition on why the margin-to-temperature mapping is approximately monotonic?
5. Would combining SMART with vector or class-wise temperature scaling bring further gains, or are these directions redundant given the per-sample mapping?
6. How would SMART behave if the calibration set is heavily class-imbalanced—does the mapping generalize across rare vs. frequent classes?

**Details Of Ethics Concerns:**

I find that some of the citations are not accurate and seem to be generated by AI. For example,
I cannot find this paper reference: Mix and match: A strategy for training object detection models with noisy annotations.
For another paper: Miao Xiong, Chien-Yi Hsieh, Beier Yang, Serim Moon, Finale Doshi-Velez, and Krzysztof Z Gajos. Proximity-informed calibration for deep neural networks. arXiv preprint arXiv:2306.04590, 2023. The author list is not correct.

---

> ### Author Response · Authors · 2025-11-25
>
> We appreciate the valuable comments from Reviewer ubNf and respond below.
>
> Re to W1:
> While SMART builds on known components, our goal is to close a practical gap under theoretical guarantees rather than offer a minor tweak. We show that the logit margin is a principled one-dimensional indicator that sharply constrains the temperature range, allowing us to replace high-capacity logit/feature networks with a lightweight margin to temperature mapping that has only a few parameters and does not scale with the number of classes. Coupled with the SmoothSoftECE objective, which is directly aligned with a smooth calibration error instead of NLL, this design addresses the tradeoff between calibrator capacity, validation size, and reliable calibration: large networks could in principle absorb many effects, but under realistic validation sizes they risk overfitting and instability. In contrast, SMART uses a simple indicator and minimal network to achieve sota calibration across architectures and datasets.
>
> Re to W2:
> For clarity, we have renamed “Huber SoftECE’’ to “Charbonnier-Smoothed SoftECE’’ (or “Smoothed SoftECE’’), reflecting our use of the Charbonnier (pseudo-Huber) loss. In the revision, SmoothSoftECE converges faster, reaches the lowest loss, and exhibits more stable, smaller gradient norms than SoftECE in Figure 3, thus preserving the calibration focus of SoftECE while offering better optimization behaviour than the original SoftECE.
>
> Re to Q1:
> The reliability diagram in Figure 1 is specific to ImageNet with ViT-B/32; its purpose is to show that different margin ranges exhibit distinct confidence–accuracy behaviour, motivating an adaptive, sample-wise treatment of margins.
>
> Re to Q2:
> Figure 2 (rightmost) is also on ImageNet with ViT-B/16, as stated in Section 3.2; we have clarified the dual y-axis layout in the revision. Early in training, optimizing NLL reduces both NLL and ECE, but after about epoch 30, NLL continues to decrease while ECE increases, illustrating the misalignment between NLL and calibration.
>
> Re to Q3:
> We use the same bandwidth λ and smoothing δ across experiments and observe that performance is quite stable under small changes in δ for reasonable λ (e.g., 0.01–0.10). The full sensitivity is shown below.
>
> **ECE**  (ImageNet, ViT-B/16) (similar patter on resnet50)
>
> | λ \ δ | 0.001 | 0.010 | 0.100 | 1.000 |
> | --- | --- | --- | --- | --- |
> | **0.01** | 1.32 | 0.84 | **0.78** | 0.85 |
> | **0.05** | 0.84 | **0.80** | 0.89 | 0.86 |
> | **0.10** | 0.99 | 0.97 | 0.81 | 2.26 |
> | **0.50** | 2.09 | 2.02 | 2.06 | 2.05 |
> | **1.00** | 2.04 | 2.48 | 2.56 | 2.56 |
>
> Similar patterns hold for CNN and ViT on ImageNet. In practice, λ=0.05 (our baseline) is a safe choice, and δ has limited impact in this regime. Values of λ≥0.5 tend to be too large, while λ=0.05 with a small δ works robustly.
>
> Re to Q4:
> We do not assume monotonicity in the margin-to-temperature mapping; it is learned from data and can vary across model–dataset pairs. On ImageNet and CIFAR-10/100 with CNNs, the mapping is almost linear and clearly increasing (Figure 16(a), r≈0.94), whereas on ImageNet with ViT-B/16 it is distinctly non-monotonic and roughly U-shaped (Figure 16(b), r≈−0.63). This suggests that margins encode uncertainty differently for CNNs and ViTs, and SMART adapts to these patterns instead of imposing a universal structure; a full theoretical explanation is left for future work.
>
> Re to Q5:
>
> | Method | ECE (%) |
> | --- | --- |
> | Uncalibrated | 5.593 |
> | VS | 2.888 |
> | CTS | 1.508 |
> | **SMART** | **0.632** |
> | VS→SMART | 1.333 |
> | CTS→SMART | 1.513 |
> | SMART→CTS | 1.999 |
> | SMART→VS | 1.955 |
>
> On ImageNet with ViT-B/16, SMART alone clearly outperforms TS, VS, and CTS, and composing SMART with VS/CTS in either order degrades calibration. This is consistent with SMART already capturing the global rescaling effects that TS/VS/CTS provide; additional global temperatures tend to undo or dilute the learned per-sample mapping. We therefore focus on SMART as a standalone calibrator.
>
> Re to Q6:
>
> Performance under Imbalance (ImageNet, ViT-B/16)
>
> | Imbalance | Class Retention | ECE(%) |
> | --- | --- | --- |
> | **Balanced** | All 100% | **0.83** |
> | **Mild** | 80%@100%, 20%@50% | 0.91 |
> | **Moderate** | 60%@100%, 40%@20% | 0.92 |
> | **Severe** | 50%@100%, 50%@10% | 0.93 |
> | **Extreme** | 30%@100%, 70%@5% | 1.01 |
>
> SMART degrades gracefully from balanced to severe imbalance, with a more noticeable drop only under the most extreme setting. For notation, 80%@100%, 20%@50% means 80% of classes: keep 100% of samples; the rest keep 50%.
>
> Ethics concerns:
> Thank you for highlighting this; we have corrected the issue in the revised version.

---

> ### Author Response · Authors · 2025-11-27
>
> Respected reviewer, should you have any further concerns, I am eagerly anticipating your response.

---

> ### Author Response · Authors · 2025-12-03
>
> This is the addition results of hyperparameter sensitivity for Q3:
>
> ECE-15 (ImageNet, ResNet50) (similar patter on vit_b_16)
>
> | λ \ δ | 0.001 | 0.010 | 0.100 | 1.000 |
> | --- | --- | --- | --- | --- |
> | 0.01 | 0.66 | 0.83 | 1.02 | 0.67 |
> | 0.05 | 0.61 | 0.66 | 0.67 | 0.66 |
> | 0.10 | 0.66 | 3.11 | 0.71 | 0.72 |
> | 0.50 | 0.85 | 0.95 | 1.29 | 1.26 |
> | 1.00 | 0.79 | 1.15 | 2.49 | 2.51 |

---

### Author Response · Authors · 2025-12-03
**Review Summary — No reply from reviewers (6,4,8,4) Before the Incident**

**Dear AC:**

Because ACs were reassigned after the recent OpenReview incident, we provide this brief summary of reviewer concerns and how they were addressed **before** the incident. No reviewers responded after our Nov 25 rebuttals, and the incident prevented further communication.

## 1. Timeline

- **Nov 25**: Rebuttal submitted with comprehensive responses.
- **Nov 27**: Follow-up comments requesting feedback.
- **Before the incident**: No reviewers replied, but all concerns were thoroughly addressed.
- **Nov 28**: OpenReview incident announced.

## 2. Reviewer Bb97 (score = 4, **primarily requesting missing benchmarks and clarifications**)

**All major concerns directly addressed with new experimental results:**

- **Missing DAC baseline** → added comprehensive DAC comparisons on ImageNet-C and ImageNet-Sketch across both ResNet50 and ViT-B/16. SMART consistently outperforms DAC variants.
- **Missing modern models** → added three requested architectures (beit_base, convnext_base, eva02_base) all showing SMART achieves best calibration.
- **Theoretical justification (Brier score)** → clarified distinction between confidence vs canonical calibration; added Brier score baseline to Table 3 showing it underperforms SMART; computed RBS metric across all benchmarks demonstrating SMART achieves best or tied-best performance.
- **Real-world distribution shift evaluation** → added WILDS iWildCam benchmark with both ID-to-ID and ID-to-OOD evaluation, both achieving lowest ECE among all methods.

**These additions directly address the reviewer's core requests for expanded baselines and evaluation scope.**

## 3. Reviewer ubNf (score = 6)

Main concerns (all addressed):

- **Incremental innovation** → clarified SMART achieves SOTA with only minimal parameters via margin-temperature theoretical insight.
- **New Loss intuition** → added Figure 3 showing improved convergence.
- **Hyperparameter sensitivity** → provided comprehensive sensitivity table showing consistency.
- **Class imbalance** → added experiments showing graceful degradation under imbalance.
- **Ethics concerns** → corrected citations.

## 4. Reviewer XnHd (score = 8)

Highly positive review (Soundness/Presentation/Contribution: all excellent). Single question about zero-shot scenarios where we clarified this is not our main target.

## 5. Reviewer 9SH3 (score = 4)

Main concerns (all addressed):

- **Novelty clarity** → clarified distinct contributions with theoretical and empirical evidence.
- **Presentation** → completely rewrote Methodology section; clarified all notation and assumptions.
- **Missing TS baselines** → added TS optimized with ECE/AdaECE via grid search, showing significant underperformance vs SMART.

---

### Note · Program_Chairs · 2026-01-17
**Submission Desk Rejected by Program Chairs**

The following references in this submission do not refer to real documents and/or have major errors in bibliographic information:

 Marcel Conde, Danny Niebling, Nicolas Schilling, and Bernhard Sick. Approaching the limit of accuracy: Residual uncertainty via test-time data augmentation. In 2023 IEEE International Conference on Data Mining, pp. 933-938. IEEE, 2023.
Arseniy Wang, Patrick Schramowski, Caner Turan, Furong Boutros, Shabab Beigpour, Bodo Rosenhahn, and Kristian Kersting. Pitfalls of in-domain uncertainty estimation and ensembling in neural networks. arXiv preprint arXiv:2302.06993, 2023.